



# Top-down and bottom-up estimates of anthropogenic methyl bromide emissions from eastern China

Haklim Choi[1], Mi-Kyung Park[1], Paul J. Fraser[2], Hyeri Park[3], Sohyeon Geum[3], Jens Mühle[4], Jooil Kim[4],
Ian Porter[5], Peter K. Salameh[4], Christina M. Harth[4], Bronwyn L. Dunse[2], Paul B. Krummel[2],
Ray F. Weiss[4], Simon O'Doherty[6], Dickon Young[6], and Sunyoung Park[1,3]

[1]Kyungpook Institute of Oceanography, Kyungpook National University, Daegu 41566, Republic of Korea
[2]Climate Science Centre, Commonwealth Scientific and Industrial Research Organisation (CSIRO) Oceans and Atmosphere,
Aspendale, Victoria 3195, Australia
[3]Department of Oceanography, Kyungpook National University, Daegu 41566, Republic of Korea
[4]Scripps Institution of Oceanography, University of California San Diego, La Jolla, California 92093, USA
[5]School of Life Sciences, La Trobe University, Bundoora, Victoria 3086, Australia
[6]Atmospheric Chemistry Research Group, University of Bristol, Bristol BS8 1TS, UK

*Correspondence to*: Sunyoung Park (sparky@knu.ac.kr)

**Abstract.** Methyl bromide ($CH_3Br$) is a potent ozone-depleting substance (ODS) that has both natural and anthropogenic sources. $CH_3Br$ has been used mainly for preplant soil fumigation, post-harvest grain and timber fumigation and structural fumigation. Most non-quarantine/pre-shipment (non-QPS) uses have been phased-out in 2005 for non-Article 5 (developed)
countries and in 2015 for Article 5 (developing) countries under the Montreal Protocol on Substances that Deplete the Ozone Layer; some uses have continued under critical use exemptions (CUEs). Under the Protocol, individual nations are required to report annual data on $CH_3Br$ production and consumption for quarantine/pre-shipment (QPS) uses, non-QPS uses and CUEs to the United Nations Environment Programme (UNEP). In this study, we analyzed high precision, *in situ* measurements of atmospheric concentrations of $CH_3Br$ obtained at the Gosan station on Jeju island, Korea, from 2008 to 2019. The background
concentrations of $CH_3Br$ in the atmosphere at Gosan declined from $8.5 \pm 0.8$ ppt in 2008 to $7.4 \pm 0.6$ ppt in 2019 at a rate of $-0.13 \pm 0.02$ ppt yr$^{-1}$. At Gosan, we also observed periods of persistent concentrations (pollution events) elevated above the decreasing background in continental air masses from China. Statistical back trajectory analyses showed that these pollution events predominantly trace back to $CH_3Br$ emissions from eastern China. Using an inter-species correlation (ISC) method with the reference trace species CFC-11 ($CCl_3F$), we estimate anthropogenic $CH_3Br$ emissions from eastern China at $4.1 \pm 1.3$ Gg
yr$^{-1}$ in 2008–2019, approximately $2.9 \pm 1.3$ Gg yr$^{-1}$ higher than the bottom-up emission estimates reported to UNEP. Possible non-fumigation $CH_3Br$ sources - rapeseed production and biomass burning – were assessed and it was found that the discrepancy is more likely due to unreported or incorrectly reported QPS and non-QPS fumigation uses. These largely-unreported anthropogenic emissions of $CH_3Br$ are confined to eastern China and account for 30–40% of anthropogenic global



CH₃Br emissions. They are likely due to delays in the introduction of CH$_3$Br alternatives, such as sulfuryl fluoride (SO$_2$F$_2$),

heat, irradiation and a possible lack of industry awareness of the need for regulation of CH$_3$Br production and use.

## 1 Introduction

Methyl bromide (CH$_3$Br) is a colorless, odorless, non-flammable chemical that is a powerful ozone-depleting substance (ODS). Apart from the issue of ozone layer depletion, there are the public health concerns associated with CH$_3$Br toxicity, for example

the risk of poisoning workers handling the chemical. CH$_3$Br has been widely used for fumigation to eradicate various pests present in soils or in the storage, import and export of grains and timbers. CH$_3$Br was listed in 1992 as an ODS under the Montreal Protocol (MP) on Substances that Deplete the Ozone Layer, an international agreement for the protection of the stratospheric ozone layer. The Parties to the Protocol agreed to a schedule for the total phase-out of CH$_3$Br use, beginning with a freeze on production and consumption in 1995 at the 1991 baseline level followed by step down reductions in 1999, 2001

and 2003 and total phase-out by 2005 for non-Article 5 (developed) countries and a freeze at 1995-1998 average baseline levels and total phase out in 2015 for Article 5 (developing) countries. Typically, 91% of CH$_3$Br consumption for non-QPS uses was for soil fumigation and 9% for storage products and structures in both non-Article 5 and Article 5 countries (MBTOC, 2018). Presently quarantine and pre-shipment (QPS) uses are exempt from the phase-out, however individual Parties to the MP are required to annually report data on the CH$_3$Br production and consumption for QPS uses, non-QPS uses and critical

use exemptions (CUEs) to United Nations Environment Programme (UNEP). Except for a small use of CH$_3$Br for CUEs, the consumption of CH$_3$Br for preplant soil fumigation and non-QPS commodities/structures has been reduced completely, contributing to an overall reduction of over 60,000 tonnes in the global consumption of CH$_3$Br. The reported amounts of consumption of CH$_3$Br for QPS that are not controlled (exempted from phase-out) under the MP have remained relatively constant over the past 20 years, and now account for more than 98% of the estimated consumption of CH$_3$Br currently reported

due to the phase-out of other uses (TEAP, 2020). Despite no formal regulation, most parties to the MP are making efforts to minimize the use of CH$_3$Br for QPS use and replace it with suitable alternatives such as heat treatment, phosphine (PH$_3$), ethyl formate (C$_2$H$_5$OCHO), sulfuryl fluoride (SO$_2$F$_2$) and ethanedinitrile (NCCN). As a consequence of this CH$_3$Br phase-out, the global atmospheric concentration of CH$_3$Br decreased from 9.2 ppt at the peak in 1996–1998, to 6.6 ppt in 2015, but then showed a slight positive growth of 0.14 ppt yr$^{-1}$ (2.1% yr$^{-1}$) from 2015 to 2016 (Engel and Rigby et al., 2019).

Since CH$_3$Br has a relatively short lifetime (0.8 year) compared to the other major ODSs (Yvon and Butler et al., 1996; Hu et al., 2012; Engel and Rigby et al., 2019), changes in surface emissions tend to be reflected quickly in changes to atmospheric concentrations. Unlike most other ODSs, CH$_3$Br has both natural and anthropogenic sources. The principal natural emission sources are the ocean (Hu et al., 2012), salt marshes (Montzka and Reimann et al., 2011), wetlands (Lee-Taylor and Holland, 2000), fungi (Lee-Taylor and Holland, 2000; Lee-Taylor et al., 2001) and plants such as mangroves or shrubs (Rhew et al.,

2001; Manley et al., 2007). Anthropogenic emission sources include fumigation (Carpenter and Reimann et al., 2014),



agricultural and biofuel biomass burning (Andreae and Merlet, 2001), and the rapeseed industry (Gan et al.,1998; Mead et al., 2008). $CH_3Br$ is removed from the atmosphere by soil and ocean deposition, reactions with hydroxyl (OH), and photolysis mainly occurring in the lower stratosphere. The sources and sinks of $CH_3Br$ in the atmosphere are not in balance, with total sinks larger than total sources by about 40 Gg yr$^{-1}$ (Carpenter and Reimann et al., 2014). This is most likely due to insufficient

understanding of existing sources (Yokouchi et al., 2002; Montzka et al., 2003). Recently, possible new sources have been identified. For example, emissions of $CH_3Br$ occur in the bread-baking process (Thornton et al., 2016) and from seaweed meadows (Weinberg et al., 2015), but their contributions were found to not have a significant impact on the global budget. The reasons for the imbalance between $CH_3Br$ sources and sinks remain unresolved.

Global anthropogenic emissions of $CH_3Br$ can be estimated using "bottom-up methods from consumption and production data

across various activities reported to UNEP annually by individual nations using activity-dependent emission factors. Significant uncertainties result from the emission factors and the speciation of $CH_3Br$ consumption across various activities (Vaughn et al., 2018). As QPS uses of $CH_3Br$ are generally highly emissive, consumption for these activities can be more accurately converted into emissions for this application (MBTOC, 2018).

"Top-down" estimates of global $CH_3Br$ emissions are derived from modelling of measured atmospheric concentrations and

atmospheric transport processes, for example using the AGAGE 12-box model of the atmosphere, assuming an atmospheric lifetime for $CH_3Br$ (Cunnold et al., 1994; Prinn et al., 2005; Rigby et al., 2013). Regional characteristics of $CH_3Br$ emissions however cannot be obtained with the AGAGE 12-box or similar model, because they do not have the resolution to account for the synoptic scale of the atmospheric flow. Since the MP control of $CH_3Br$ consumption applies at a national level, rather than globally, it is important to estimate the resultant top-down emissions at a regional to national scale (Weiss and Prinn, 2011).

China is the largest producer and consumer of agricultural products in the world and therefore has potentially large anthropogenic sources of $CH_3Br$ and is an important region for understanding $CH_3Br$ emissions in East Asia. Several studies have estimated the regional or national emissions from China based on "top-down" approaches using atmospheric observations. Blake et al. (2003) estimated the $CH_3Br$ emissions of 2.6 Gg yr$^{-1}$ in China (South China: 2.0 Gg yr$^{-1}$ and North China: 0.6 Gg yr$^{-1}$) from aircraft observations in 2001. An inverse modeling study (Vollmer et al., 2009) using high-frequency ground

measurements suggested emissions from China had decreased to 0.24 Gg yr$^{-1}$ in 2006–2008. However, those results were based on a limited period of observations (e.g., few months to years) and could not analyze the long-term variations and trends in $CH_3Br$ emissions. Since then, there have been no further studies tracking the $CH_3Br$ emission trends in East Asia.

In this study, we present the 12-year high-precision, high-frequency record of atmospheric $CH_3Br$ concentrations observed at Gosan station on Jeju island, South Korea, and analyze the observed variations in atmospheric $CH_3Br$. We estimate annual

emissions of $CH_3Br$ mainly from anthropogenic sources in eastern China, based on the empirical inter-species correlations between $CH_3Br$ and CFC-11 during pollution episodes from eastern China and the well-defined eastern China CFC-11 emissions. This is the first study to present the long-term changes in $CH_3Br$ emissions from eastern China, after the phase-out period. In the following sections, we first introduce (Section 2) the Gosan station and the *in situ* ground-based instrumentation for $CH_3Br$ measurements in Section 2. In Section 3, long-term seasonal and annual variations of atmospheric $CH_3Br$



concentrations are discussed. In Sections 4 and 5, we suggest potential source regions that show high sensitivities to the enhanced CH₃Br concentrations based on air-mass back-trajectory statistics. The observation-based emission estimates of CH₃Br in eastern China are further discussed considering the existing discrepancy between the global bottom-up and top-down emissions of CH₃Br.

## 2 Instrumentation

The coastal atmospheric observation station Gosan (GSN, 33.3°N, 126.2°E, 72 m a.s.l) at the south-western tip of Jeju island, South Korea (See Figure 1) is ideally located to monitor regional background concentrations of atmospheric trace gases due to minimal influence of local anthropogenic pollution sources, and the strong pollution outflows from China, Korea, and Japan in East Asia (Kim et al.2012; Li et al., 2011 and 2014; Park et al., 2018).

The *in situ* measurement system at Gosan, a "Medusa" gas chromatography-mass spectrometer (GC-MS) equipped with a
cryogenic pre-concentration system (Miller et al., 2008; Prinn et al., 2018), monitors more than 40 halogenated compounds including CFC-11 and CH₃Br. As a part of the Advanced Global Atmospheric Gases Experiment (AGAGE; Prinn et al., 2018), Gosan station has been conducting continuous high-precision and high-frequency observations approximately every 2-hours (12 times per day) from 2008 to the present. The precisions ($1\sigma$) of all species, determined from repeated analysis (*n*=12) of an ambient standard, are better than 1% (i.e., the precision of CH₃Br < 0.1%). The atmospheric abundances of most the Medusa
compounds are calibrated on scales maintained by the Scripps Institution of Oceanography (SIO) (e.g., SIO-05 scale for CH₃Br in this study).

## 3 Description of measurement data

Long-term, high-frequency CH₃Br data observed during 2008–2019 at Gosan and background concentration data from Mace Head (53.3°N, 9.9°W) and Cape Grim (40.7°S, 144.7°E) are shown in Figure 2. Regional background concentrations of CH₃Br
were determined by removing pollution events after applying a polynomial fit to the lower 99.7% (within $3\sigma$) of the Gaussian distribution derived from the 121-day observations for 60 days before and after each observed data point (O'Doherty et al., 2001). The baseline concentrations at Gosan and Mace Head (northern hemisphere) are higher than those of Cape Grim (southern hemisphere), and while the annual cycles at Gosan and Mace Head are similar.

The annual average CH₃Br baseline concentrations decreased steadily from 8.5 ± 0.8 ppt in 2008 to 7.4 ± 0.6 ppt in 2019
(Table 1), declining at a rate of -0.13 ± 0.02 ppt yr⁻¹ (-1.5% yr⁻¹). This rate of decline for CH₃Br is consistent with the global trend of atmospheric CH₃Br determined from AGAGE *in situ* and NOAA (National Oceanic and Atmospheric Administration) flask data in 2011–2012 (Carpenter and Reimann et al., 2014), which has been attributed to the influence of the CH₃Br restrictions on non-QPS use imposed by the Montreal Protocol.





The monthly mean CH$_3$Br baseline concentrations for 2008–2019 are shown in Figure 3. The seasonal variations show a steady

increase in spring, reaching a maximum in May, then dropping in June-July, followed by a constant level for the last 5 months of the year. The various sources and sinks of CH$_3$Br likely show seasonal variability, and the summertime minima in CH$_3$Br can be largely explained by the atmospheric concentrations of OH reaching a maximum during the boreal summer (Cox, 2002; Simmonds et al., 2004) and long-range transport of southern hemispheric air parcel that over-cross the tropical regions (Li et al., 2018).

Despite the continuous decease in background concentrations, we observed clear pollution signals (shown in red in Figure 2) through the entire study period, representing persistent inflow to Gosan of air masses influenced by regional CH$_3$Br emission sources and thus containing elevated concentrations of CH$_3$Br. The annual means of the enhancement concentrations (pollution – baseline) are consistently in a range of 3.5 to 5.0 ppt as given in Figure 4. Note that there are data missing for several months in 2016, 2017, and 2018, mainly because of typhoon damage to the Gosan station in early summer/fall.

**4 Potential CH$_3$Br source regions**

The regional distribution of potential CH$_3$Br sources in East Asia was derived by applying a statistical analysis of back trajectories corresponding to the observed CH$_3$Br enhancements at Gosan from 2008 to 2019. Air mass back trajectories were generated using the Hybrid Single-Particle Lagrangian Integrated Trajectory (HYSPLIT; Draxler et al., 1998) model from the NOAA Air Resources Laboratory with meteorological data output from the Global Data Assimilation System (1˚×1˚ horizontal

resolution, 23 vertical layers, ftp://arlftp.arlhq.noaa.gov/pub/archives/gdas1). The HYSPLIT 6-day air mass backward trajectories were initialized 500 m above the Gosan observation station, at a height where topographical influences can be minimized (Li et al., 2014). To minimize the error that arises from a small number of outlier trajectories, only grids with more than 12 over-passing trajectories were used to define a potential source region (Reimann et al., 2004; method described in SI). Figure 5 shows the distribution of potential CH$_3$Br source regions in East Asia for 2008–2019, widely distributed over eastern

China and southern Korea. In particular, high potential source regions for CH$_3$Br emissions are seen along the Yangtze River that connects Shanghai, Nanjing, Hefei and Wuhan. The port of Shanghai is one of the busiest container ports in the world since 2010, with high volumes of port traffic and a large population (Robert et al., 2020). For example, Shanghai handled 43.3 million twenty-foot equivalent units in 2019 (https://safety4sea.com/port-of-shanghai-worlds-busiest-container-port-for-2019/, last access: 11 March, 2021). In several cases, high CH$_3$Br concentrations were observed in narrow-width air mass back

trajectories that showed long residence times over the port of Shanghai (see Figure S1), which would be consistent with Shanghai being a likely major port for QPS usage of CH$_3$Br.

The high potential source regions include not only modern industrial urban areas but also the vast alluvial plains along the Yangtze River and its main tributaries. Note that this statistical analysis has little sensitivity to emissions from southwestern-western China and tends to over-estimate source strengths near the boundary due to the limits of the 5–6 day backward

trajectory domain of the HYSPLIT model. Therefore, those parts of China have been excluded from further discussion (Park





et al., 2018). Also note that this statistical trajectory analysis tends to underestimate emissions at sub-grid scale hotspots because the measured concentration gets distributed evenly over the grid cell (Stohl, 1996). Also, the dilution effects on distant source emissions are not considered in this statistical approach (Vollmer et al., 2006). Thereby, the emissions from nearby sources might be overestimated due to the higher CH₃Br concentration. For this reason, the emission potential for South Korea,

shown in Figure 5, may be lower. We do not attempt to identify more exact locations of CH₃Br emission sources based on this approach because of its potential uncertainties, nevertheless it is clear that significant emissions of CH₃Br originate predominantly from eastern-central China.

## 5 Estimation of CH₃Br Emissions from eastern China using an inter-species correlation method

In the previous section, it was noted that most of the air masses exhibiting enhanced CH₃Br concentrations flow into Gosan
from eastern China. We classify the air mass origins into 17 regions (see Figure S2a for the regional domains) based on the 6-day kinematic back trajectories of the HYSPLIT model. If a trajectory arriving at Gosan had entered the boundary layer (as defined by HYSPLIT) only within the regional domains for eastern China-1 (region 16), eastern China-2 (region 5), and Shandong provinces (region 9), it was defined as an air mass originating from eastern China. The air mass classification applied to the CH₃Br time series is illustrated in Figure S2b. The proportions of CH₃Br pollution events from 2008 to 2019 classified
into China, South Korea, and other regions were 37, 44, and 19%, respectively. Among them, 98% of air masses classified as China correspond to eastern China (~35% of the total).

### 5.1 Interspecies correlation method

To estimate emission of CH₃Br from eastern China, we applied an interspecies correlation (ISC) method (Palmer et al., 2003; Dunse et al., 2005; Yokouchi et al., 2006; Millet et al., 2009; Li et al., 2011; Wang et al., 2014; Park et al., 2018). Described
as a "ratio-method", this approach can derive the emission of a trace gas of interest from the correlation of its enhancement above baseline with that of a reference compound. This empirical ratio approach can estimate regional emissions of various substances in a simple and robust manner, compared to inverse methods that require complex computational processes in combination with chemical transport models. For a reference tracer in ISC method, the following conditions are required: i) long lifetime, thus low chemical reactivity during transport from source to observation site, ii) well-quantified emission sources,
iii) approximate co-location of the source regions for the reference and target species resulting in significant correlations with the target species. Several previous studies have used carbon monoxide (CO) as a reference species for ISC (Palmer et al, 2003; Dunse et al., 2005; Guo et al., 2009; Wang et al., 2014). CO can be observed readily at the target species observation sites and often has documented emissions from anthropogenic sources, usually as a component of regional air quality emissions inventories. One of the issues of using CO as a reference species in ISC is that the emission inventories usually document
anthropogenic CO sources only, whereas the observations see CO emissions from anthropogenic and natural sources, biomass



burning for example. Therefore, in the CO observational data record, CO pollution episodes have to be identified as predominantly anthropogenic before inclusion in the ISC emissions calculations, which complicates matters.

Instead, we selected CFC-11 as the reference compound, because CFC-11 has a long lifetime (50–60 years) with low chemical reactivity, has been measured simultaneously with CH₃Br at Gosan showing strong correlations (Li et al., 2011), and the CFC-
11 emissions from eastern China for 2008–2019 have been very well-quantified by four different inverse models (Rigby et al., 2019; Park et al. 2021).

Atmospheric observations of CH₃Br, CFC-11, benzene, toluene and ethane (Figure 6), show significant correlations between CH₃Br and CFC-11, for example on 19 and 21 May. The volatile organic compounds (VOCs), like benzene, toluene and ethane, are emitted from biomass burning, show a noteworthy simultaneous increment, suggesting that biomass burning in eastern
China could be a potential CH₃Br source. This point is discussed in detail in the next session.

The emissions of CFC-11 are from anthropogenic sources only – there are no natural sources of CFC-11. Although the emission sources of CFC-11 and CH₃Br are not necessarily co-located on an emission activity basis, we can still apply ISC method to estimate the magnitude of country/regional scale emissions of CH₃Br, when they occur within a same country/region where CFC-11 is emitted. When the likely CFC-11 and CH₃Br sources are not co-located on a fine scale but are co-located on a
regional scale, then it is important to make the CFC-11 and CH₃Br observations sufficiently distant (hundreds of km) from the source region so that the initial individual plumes of CFC-11 and CH₃Br emissions from separate sources become well mixed. In this study, the emissions of CH₃Br in eastern China are derived using the following equation:

$$E_{MB} = E_{CFC-11} \times \alpha \times \frac{M_{MB}}{M_{CFC-11}}, \tag{1}$$


where, $E_{MB}$ and $E_{CFC-11}$ are the emissions of CH₃Br and CFC-11, respectively, $\alpha$ is a slope of linear regression between enhancements of CH₃Br and CFC-11 (ΔCH₃Br and ΔCFC-11), $M_{MB}$ and $M_{CFC-11}$ are the molecular weights of CH₃Br and CFC-11, respectively. The intercept term of the linear regression can be ignored because it is generally not significantly different than zero, confirmed by the similar slope terms from linear and linear-through-the-origin regressions (Dunse et al.,
215   2005).

The uncertainty of CH₃Br emissions is associated with uncertainties of $\alpha$ and $E_{CFC-11}$ and determined by an error propagation method as follows:

$$\sigma E_{MB} = \sigma E_{CFC-11} + \sigma\alpha \tag{2}$$


where, $\sigma E_{MB}$ is the uncertainty of estimated CH₃Br emissions, $\sigma E_{CFC-11}$ and $\sigma_\alpha$ are the uncertainties of $E_{CFC-11}$ and $\alpha$, respectively.





## 5.2 CFC-11 emissions from inverse model frameworks

We use known emission estimates of CFC-11 from eastern China, which were derived by inverse modelling of Gosan CFC-
11 observation data (Rigby et al., 2019; Park et al., 2021). Atmospheric concentrations for CFC-11 observed over the same period with CH$_3$Br are shown in Figure S3. CFC-11 emissions were estimated from four different Bayesian inverse methods based on two different Lagrangian atmospheric chemical transport models: the UK Met Office Numerical Atmospheric-dispersion Modelling Environment (NAME; Jones et al., 2007) and the FLEXible PARTicle dispersion model (FLEXPART; Stohl et al., 2005; Pisso et al., 2019). The details for the modelling frameworks are described in Rigby et al. (2019) and Park
et al. (2021). For $E_{CFC-11}$ and $\sigma_{CFC-11}$ in Eq. 1 and 2, the emissions of CFC-11 and their uncertainties are derived from the four inversion models used (Park et al., 2021). The average value of estimated emissions of CFC-11 for eastern China ranged from 5.7 to 20.4 Gg yr$^{-1}$ over the period 2008–2019 (Park et al., 2021).

## 5.3 Linear regression

Several studies have used ordinary least squares (OLS) as a linear regression method due to its simplicity. With OLS, the errors
of both independent variables are not considered. However, if both variables have uncertainties like the observation data used in this study, both errors must be considered when performing a linear regression between the two variables. Some other linear regression methods considering the XY errors have been suggested to overcome the limitation of OLS. Dunse et al. (2005) used the Fitexy method (Press et al., 2007), Wang et al. (2014) used the Orthogonal Distance regression (ODR; Wallace et al., 2012), and Park et al. (2018) used the Williamson-York regression (WYR; Cantrell, 2008) to estimate the emissions of the
trace gases by ISC method. A recent study (Wu and Yu, 2018) suggested that Weighted Deming Regression (WDR; hereafter, DR) method estimates a relatively more accurate slope and intercept by minimizing the residual errors both X and Y among the various linear regression methods, particularly for atmospheric data with measurement error. As mentioned earlier, the calculated slopes can be different depending on which linear regression fit is used. Therefore, we applied not only the DR approach but also the Fitexy and WYR methods to determine annual slopes between the observed enhancements of CH$_3$Br and
CFC-11 during 2008 to 2019. The results for the Fitexy and WYR methods are similar. Even though the co-matched observation points were slightly scattered in the range of large enhancements, the DR generated best fits representing the overall correlations trends. Millet et al. (2009) required a Pearson correlation coefficient (R) over 0.3. Figure S4 shows the resulting annual slopes. For most of entire observation period, CH$_3$Br enhancements show a significant correlation with those for CFC-11 with R larger than 0.4 (e.g., R = 0.7 in 2009). They do not maintain a high correlation (R < 0.4) for every single
year since most of the enhancements of CH$_3$Br and CFC-11 were less than 5 ppt and high pollution events occurred only occasionally within a year. Nevertheless, CFC-11 seems suitable as a reference compound to trace anthropogenic emissions from eastern China.



### 5.4 Estimation of CH₃Br emissions from eastern China

Figure 7 shows the annual $CH_3Br$ emissions estimates derived for eastern China by ISC method from atmospheric
measurements at Gosan from 2008 to 2019. The bar plots represent annual $CH_3Br$ emissions with 1-σ uncertainties, which
were determined based on CFC-11 emissions derived from four different inversion frameworks. The results derived from
different inversion methods agree within the stated uncertainties for most years. Note that the emissions estimate in 2013
calculated from the NAME-HB CFC-11 inversion was 2.6 Gg yr$^{-1}$, while those from other inversions were larger than 5 Gg
yr$^{-1}$. Despite the uncertainty ranges for the CFC-11 inversion results and for the least-squares fits in the ISC method, the
resulting $CH_3Br$ emissions from eastern China have remained relatively constant at 4.1 ± 1.3 Gg yr$^{-1}$ for the period 2008–2019,
with small fluctuations from year to year. This represents 40–50% of the summed global emissions of $CH_3Br$ for QPS (on
average of 8.0 Gg yr$^{-1}$) and non-QPS (on average of 2.2 Gg yr$^{-1}$) fumigation usage in 2008–2019 (see Table S2; Carpenter and
Reimann et al., 2014; TEAP, 2020).

The emissions of $CH_3Br$ peaked in 2010 at 7.1 ± 1.3 Gg yr$^{-1}$ and then decreased to 2.4 ± 1.3 ppt in 2012, followed by a slight
increasing trend in later years. The abrupt increase of $CH_3Br$ emissions in 2010 is difficult to explain in terms of the
consumption and production data reported to the Ozone Secretariat for both controlled uses and QPS uses of $CH_3Br$. The
consumption in 2010 and 2012 was less than 1.5 Gg yr$^{-1}$ and possible emissions between 1.0 – 1.4 Gg yr$^{-1}$. The cause(s) of the
relatively large emissions in 2010 and 2013 are unknown. One possibility is wildfire $CH_3Br$ emissions from China modulated
by the El Niño-Southern Oscillation (ENSO). South-western China showed an ENSO-related maximum in fire occurrences in
2010 and south-eastern China in 2013 (Fang et al., 2021). The increase in $CH_3Br$ emissions for 2014–2018 possibly reflects
the impact of increased QPS $CH_3Br$ use in traded commodities as reported to UNEP (MBTOC, 2018).

Figure 8 shows the comparison between bottom-up emissions of $CH_3Br$ for China reported to UNEP and top-down emissions
of $CH_3Br$ derived by ISC for eastern China using CFC-11 as the reference emissions. The detailed values of each category
are described in Table 2. The bottom-up emissions of $CH_3Br$ used in fumigation are determined by applying an emission factor
of 65% to the reported non-QPS consumption and 84% to the reported QPS consumption (MBTOC, 2006).

As mentioned earlier, the increase in bottom-up emissions of $CH_3Br$ over the period 2014–2018 is consistent with an increase
in consumption for QPS fumigation. However, in 2019, the reported non-QPS and QPS consumptions were reduced to zero
and 0.87 Gg yr$^{-1}$, respectively. The total calculated emissions of $CH_3Br$ in 2019 (0.73 Gg yr$^{-1}$) are lower than in 2008 (1.44 Gg
yr$^{-1}$). The average of the bottom-up emissions of $CH_3Br$ from China is 1.1 ± 0.2 Gg yr$^{-1}$ in the period of 2008–2019.

### 5.5 Potential of anthropogenic sources that contribute to CH₃Br emissions

Overall, the variations of both bottom-up and top-down emissions exhibit qualitative agreement, with peak emissions in 2010,
a decrease in 2011 and 2012 and a slight increase until 2017 and 2018 (except for the large top-down emissions in 2010 and
2013 discussed above), and then decreasing again in 2019. However, there is an obvious, significant discrepancy between the
absolute values of both data sets. Considering the bottom-up emissions were based on reported data for all of China and the





top-down emissions were derived for eastern China, actual difference in derived emissions of $CH_3Br$ is likely to be larger. Assuming that the emissions from eastern China represent the entire Chinese emissions. the mean difference between the bottom-up and top-down estimates over the entire period 2008–2019 is $2.9 \pm 1.3$ Gg yr$^{-1}$. The largest difference was in 2010 (5.8 Gg yr$^{-1}$), with top-down emissions (7.1 Gg) nearly a factor of 6 times greater than the bottom-up emissions (1.2 Gg). The causes of these large discrepancies in estimated emissions of $CH_3Br$ are not obvious.

We have examined some possibilities –

(i) Rapeseed industry

In the life cycle of rapeseed, $CH_3Br$ is largely emitted during the flowering period in the 2 months after sowing (Jiao et al., 2020). Rapeseed in the northern hemisphere generally blooms in the warm weather from March to May. So seasonal emissions from the arable land of rapeseed may be related to the observed springtime increase in $CH_3Br$ polluted concentrations at Gosan

(See Figure S5).

China is the third-largest producer of rapeseed in the world after the European Union and Canada, accounting for 12% of the total rapeseed production in 2015–2016, and the arable land lies mainly along the Yangtze River, which is suitable for growing rapeseed (Khir et al., 2017). Previous studies have reported that the global emissions of $CH_3Br$ by the rapeseed industry range from $2.8 \pm 0.7$ Gg yr$^{-1}$ (Jiao et al., 2020) to 5 Gg yr$^{-1}$ (Gan et al., 1998; Mead et al., 2008). Considering the proportion of eastern

China in global rapeseed industry, the emissions of $CH_3Br$ by rapeseed in eastern China could be about 0.3–0.6 Gg yr$^{-1}$.

(ii) Biomass burning

Owing to the almost total phase out of $CH_3Br$ for non-QPS uses to date, the largest contributor to global anthropogenic emissions of $CH_3Br$ is biomass burning, such as agricultural open-field burning and use of biofuels (about 23 Gg yr$^{-1}$; Carpenter and Reimann et al., 2014). As shown in Figure 6, the elevated concentrations of VOCs (toluene, benzene, ethane), which are

associated with biomass burning, are correlated with elevated concentrations of $CH_3Br$, suggesting that there may be some contribution of biomass burning to the observed $CH_3Br$ enhancements. Note, the sources of VOCs pollution are generally not entirely due to biomass burning, as VOCs are emitted by combustion processes in general (e.g., fossil fuel use and combustion). Approximately 140 Tg of agricultural residues are burned in fields across all of China every year (Zhao et al., 2017). Biomass burning in eastern China is predominantly due to the burning of agricultural crop residues (~60 Tg yr$^{-1}$), mainly wheat residues

(in May–June), rice and corn residues (in September–October) (Zhang et al., 2020). This eastern China biomass burning seasonality may contribute partly to the seasonality in elevated levels of $CH_3Br$ seen at Gosan (May–June and September– October, see Figure S5).

The global annual emissions of $CH_3Br$ from the burning of agricultural waste are uncertain. Recently, Andreae (2019) has revised the emission factor (EF) of $CH_3Br$ by agricultural residues from the field experiment to 1.1 g tonnes$^{-1}$, and based on

this, the global biomass burning emission of $CH_3Br$ by agricultural residues estimates was 0.3 Gg yr$^{-1}$. Using this EF, the emissions of $CH_3Br$ from biomass burning of agricultural residues in eastern China would be approximately 0.07 Gg yr$^{-1}$.

(iii) Post-harvest treatment


Historically, $CH_3Br$ consumption results from soil fumigation (non-QPS), structural fumigation (non-QPS) and post-harvest fumigation (mainly QPS). Currently, the phase-out of $CH_3Br$ has been successfully implemented under the Montreal Protocol for non-QPS applications, in particular the decrease in consumption of $CH_3Br$ for soil fumigation. Chemicals (e.g., chloropicrin, metam sodium, dazomet, etc.) and non-chemical methods (steam, soilless culture, resistant varieties) have been successfully introduced as alternatives to $CH_3Br$ use as soil fumigants (Mao et al., 2016; MBTOC, 2018). For QPS applications, phosphine has been widely used as a substitute for $CH_3Br$ in post-treatment of commodities, but it is known that some pests have developed resistance to phosphine (Jagadeesan and Nayak, 2017; Xinyi et al., 2017). $SO_2F_2$ is used in China as an alternative to non-QPS use of $CH_3Br$ for the pre-plant soil fumigation as well as the QPS disinfestation of some durable products and post-harvest commodities (Cao et al., 2014; Gressent et al., 2021). Interestingly, the spatial distribution of the potential emission source regions estimated from the $SO_2F_2$ pollution observed at Gosan is very similar to that for $CH_3Br$ (Figure S6). In addition, the concentrations of $SO_2F_2$ and $CH_3Br$ increase contemporaneously, and the correlations between the enhancements of both substances and CFC-11 are significant (Figure S7 and S8). This implies temporal and spatial co-emissions of $SO_2F_2$ with anthropogenic $CH_3Br$ into the atmosphere. Gressent et al. (2021) showed that $SO_2F_2$ emissions in China were predominantly generated by post-harvest treatment rather than structural fumigation and were distributed within a large portion in East China. Using these Gressent et al. $SO_2F_2$ emissions, the $CH_3Br$ emissions from eastern China for post-harvest treatment derived by the ISC method from the observations of $SO_2F_2$ and $CH_3Br$ at Gosan were $0.9 \pm 0.2$ Gg yr$^{-1}$ for the period 2014–2019. Thus, the post-harvest use of $CH_3Br$ in eastern China results in approximately 1 Gg yr$^{-1}$ of anthropogenic $CH_3Br$ emissions.

(iv) Unreported or inaccurately reported emissions from fumigation usage

The $CH_3Br$ emissions proposed above in (i)–(iii) can account for about half of the discrepancy (2.9 Gg yr$^{-1}$) between 'top down' and 'bottom up' estimates for east China. The sources of the remaining discrepancies (~1.4 Gg yr$^{-1}$) in $CH_3Br$ emissions remain unknown.

Errors in the reported inventory for regulated uses cannot be ruled out because it is unsure whether the limits on new QPS use have been adhered to (MBTOC, 2018). Besides, despite the successful reduction of anthropogenic $CH_3Br$ emissions globally, the possibility of unidentified sources of emissions has been raised in multi-year MBTOC assessment reports (Porter and Fraser, 2020). As a similar example, we note that, although CFC-11 was a very important target chemical for phase-out under the Montreal Protocol, unexpected CFC-11 emission increases were found due to unreported production and use in eastern China during 2013–2018 (Rigby et al., 2019; Park et al., 2021). In addition, it may be premature to conclude that $CH_3Br$ non-QPS use in China has been completely replaced by the alternatives discussed above. Since $CH_3Br$ represents the most effective and the cheapest fumigation method, the transition to the use of alternatives may be delayed without strong regulations and/or financial incentives and/or social awareness. The fact that $CH_3Br$ emissions derived from atmospheric observations in this study are significantly larger than reported emissions suggests that unreported fumigation use of $CH_3Br$ may have occurred during the transition to alternative fumigation methods or that other sources, such as emissions from industrial wastes, have been overlooked.



## 6 Summary and conclusion

Atmospheric $CH_3Br$ has both natural and anthropogenic sources and plays a significant role in stratospheric ozone destruction.
For this reason, $CH_3Br$ non-QPS uses as a soil, commodity treatment and structural fumigant are being phased-out globally
under the Montreal Protocol on Substances that Deplete the Ozone Layer, and its QPS use as a commodity fumigant is regulated.
To understand the temporal trend in atmospheric $CH_3Br$ abundances and its emission sources in East Asia, we analyzed the
concentrations of $CH_3Br$ observed at Gosan (Jeju Island, South Korea) for 12 years from 2008 to 2019. The baseline
concentrations indicating the regional state of the background atmosphere have decreased by $-0.13 \pm 0.02$ ppt $yr^{-1}$ ($-1.5$ % $yr^{-1}$) during the period, with seasonal variations increasing in spring and decreasing in summer. Despite the decreasing trend of
the $CH_3Br$ baseline, relatively constant-strength pollution events occurred in every year.

A statistical backward trajectory analysis showed that emissions of $CH_3Br$ in the region were highest from eastern China
compared to other surrounding countries. Top-down emissions estimates of $CH_3Br$ from eastern China were determined by
using an ISC method with CFC-11 as the reference tracer defining anthropogenic $CH_3Br$ emissions. The ISC-based $CH_3Br$
emission rates were $4.1 \pm 1.3$ Gg $yr^{-1}$ on average during 2008–2019 and, despite the $CH_3Br$ phase-out for non-QPS applications
in Article 5 countries, which includes China, in 2015, significant $CH_3Br$ emissions have continued. These $CH_3Br$ emissions
determined from atmospheric observations are significantly different from the bottom-up emission estimates predicted from
consumption data reported to UNEP ($1.1 \pm 0.2$ Gg $yr^{-1}$). The possible contributions of rapeseed production and biomass burning
to this discrepancy were assessed at approximately $0.6 \pm 0.2$ Gg $yr^{-1}$), insufficient to explain the approximate 3 Gg $yr^{-1}$
difference between top-down ($4.1$ Gg $yr^{-1}$) and bottom-up ($1.1$ Gg $yr^{-1}$) estimates.

The remaining discrepancy ($3.5$ Gg $yr^{-1}$) is most likely due to fumigation use that was not reported and/or inaccurately reported
or emissions from unknown sources, such as industrial waste or other sources. Correlations between $CH_3Br$ and $SO_2F_2$
pollution levels at Gosan suggest that the post-harvest use of $CH_3Br$ in eastern China contributes $0.9 \pm 0.2$ Gg $yr^{-1}$ to this 3.5
Gg $yr^{-1}$ discrepancy. These data may suggest that the transition from $CH_3Br$ to $SO_2F_2$ or other alternatives for post-harvest
fumigation in eastern China is only partially complete. Unreported use for fumigation may be related to the delay in introducing
alternative technologies to $CH_3Br$ fumigation in east China and/or the lack of social awareness of the regulation, during the
transitional period to alternative technologies.

Most of our estimated emissions of $CH_3Br$ are from eastern China and these $CH_3Br$ emissions, likely from unreported or
inaccurately reported fumigation usage, are significant enough to account for 30–40% of global emissions for fumigation usage.
Further analysis of $CH_3Br$ emissions from all of China would enhance understanding of these potentially
unreported/underestimated emissions. Our method has limitations in considering all sources of $CH_3Br$ and thus inherent
uncertainties. Nevertheless, it is important to investigate the accuracy of bottom-up emission inventories for anthropogenic
sources of $CH_3Br$ using comparisons with observation-derived top-down emissions estimates as presented here.





The total tropospheric bromine (in units of ppt) from long-lived brominated substances (CH$_3$Br and halons) controlled by the
385 MP has been decreasing since reaching a peak in 1998, mainly due to the decline of CH$_3$Br. However, the contributions of halons to declining tropospheric bromine have become predominant since 2012 (Carpenter and Reimann et al., 2014). In recent years, CH$_3$Br accounts for a significant proportion of the total amount of bromine in the troposphere from long-lived compounds. Consequently, if any unreported non-QPS and QPS emissions from fumigation usage could be reduced and eventually stopped in developing countries, a further reduction of atmospheric CH$_3$Br concentrations would occur very quickly,
390 due to the short half-life of CH$_3$Br. For this reason, continued monitoring of atmospheric CH$_3$Br concentrations in East Asia and improvements in inverse modelling approaches are presently seen as a key priority in order to locate and identify specific emission sources.

**Data availability**

Data used in this study are available from the AGAGE (Advanced Global Atmospheric Gases Experiment) database
395 (http://agage.eas.gatech.edu/data_archive/agage/gc-ms-medusa/, last access: August 2021)

**Author contributions**

HC, SP, and PJF designed the study; HC, SP, PJF, IP, JM, and JK interpreted the analyzed results and wrote the manuscript; HC, SP, MP, HP, and SG carried out the measurement of CH$_3$Br and CFC-11 at Gosan; JM, PKS, CMH and RFW supported the calibration and long-term precision for the observations at Gosan; SOD and DY provided the *in situ* measurement data
400 from Mace Head; PJF, BLD, and PBK provided the *in situ* measurement data from Cape Grim.

**Competing interests**

The authors declare that they have no conflict of interest.

**Acknowledgements**

This research was supported by the National Research Foundation of Korea (NRF) grant funded by the Korean government
405 (MSIT) (no. 2020R1A2C3003774). Support for contributions by J. Kim, J. Mühle, C. M. Harth, P. K. Salameh, and R. F. Weiss came from National Aeronautics and Space Administration (grant nos. NNX16AC96G and NNX16AC97G). Support for contributions by P. J. Fraser, B. L. Dunse, and P. B. Krummel came from National Aeronautics and Space Administration (grant no. NNX16AC98G), the Australian Bureau of Meteorology, CSIRO, the Australian Department of Agriculture, Water and the Environment (DAWE). Support for contributions by I. Porter funded by La Trobe University.





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





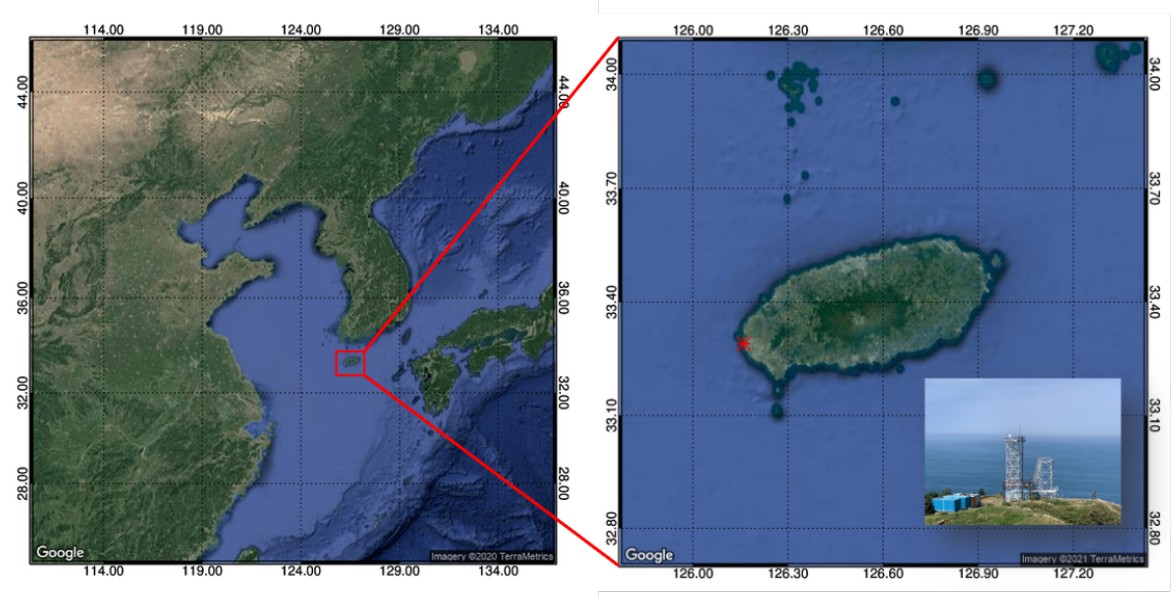

**Figure 1: Gosan station (33.3° N, 126.9° E, 72 m a.s.l.) on Jeju island, Korea (red asterisk). Air samples are taken at 17 m (~100 m a.s.l.) from a tower next to the coastal cliff. (Map data: © Google Earth)**





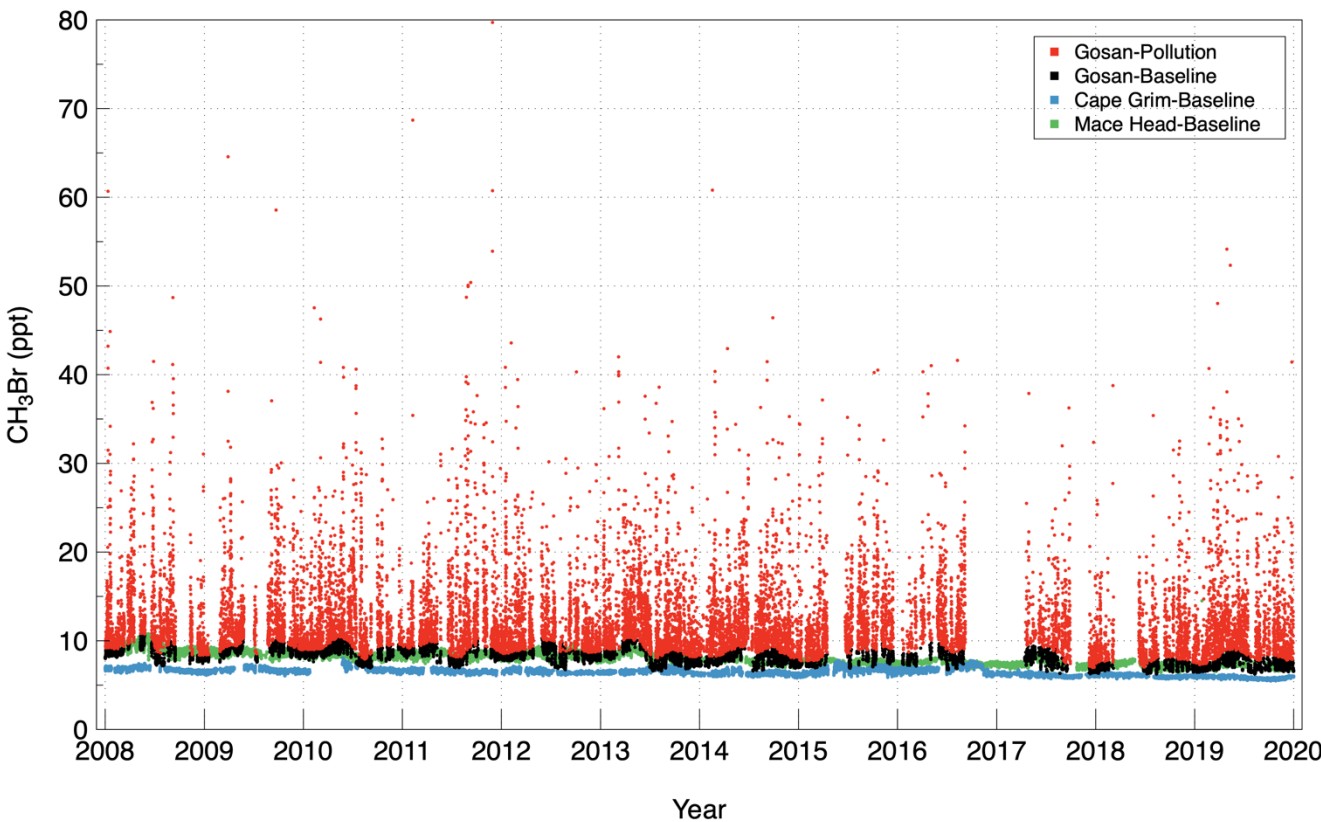

**Figure 2: Concentrations of CH₃Br in the atmosphere at Gosan for the period 2008–2019. The baseline data (black) are selected using a statistical method (O'Doherty et al., 2001); the polluted data (red) are elevated above the baseline data. The baseline data from Mace Head, Ireland (green) and Cape Grim, Australia (blue) over the same period are shown as mid-latitude references for the Northern and Southern Hemisphere, respectively.**






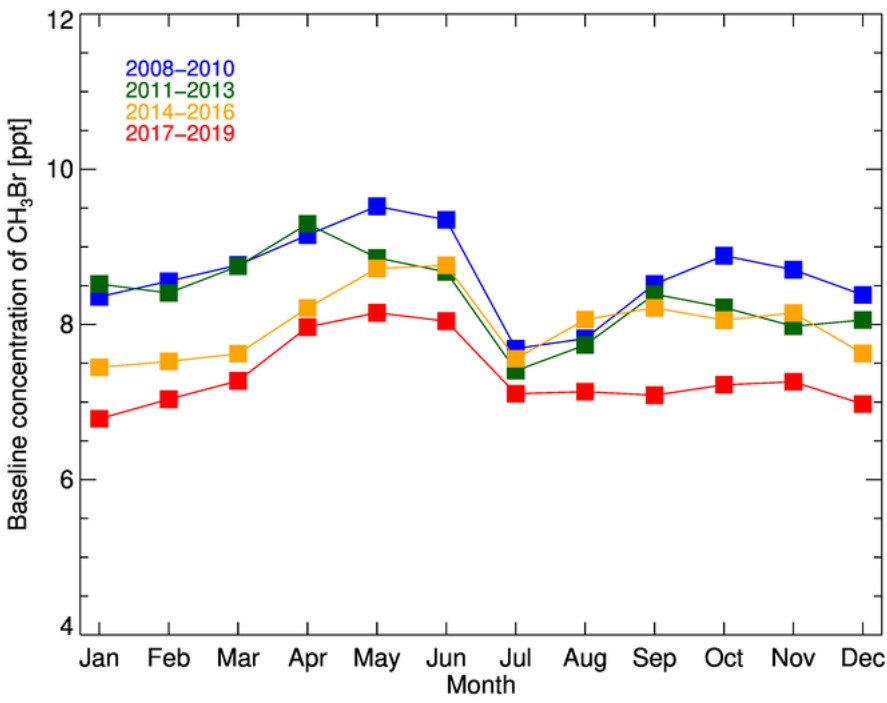

**Figure 3: Monthly mean CH₃Br baseline concentrations at Gosan for 2008–2019. Each color represents the average of the 3-year interval for the period. Note that there are data missing for several months in 2016, 2017, and 2018, mainly due to typhoon damage to Gosan station.**





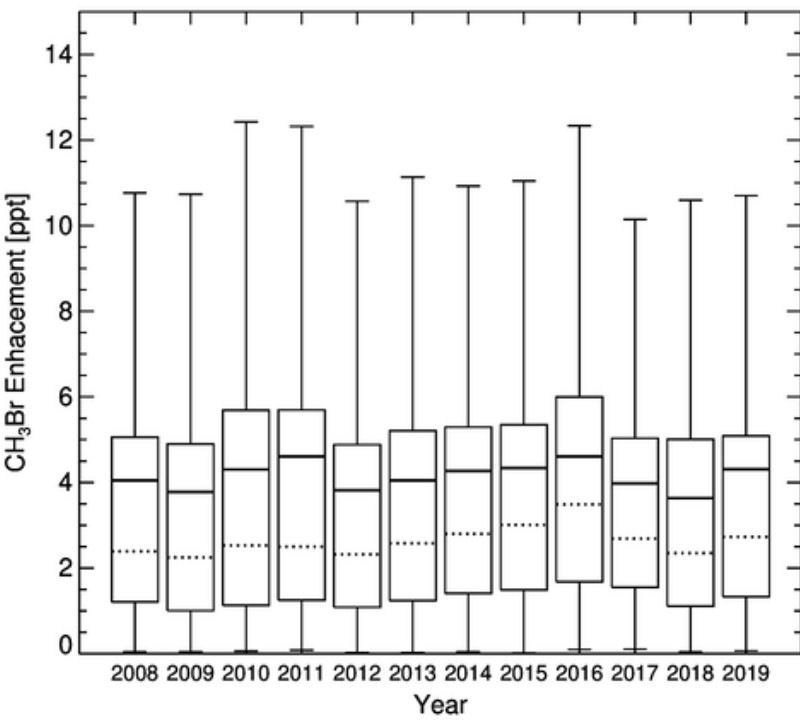

**Figure 4: Box-whisker plot of annual enhancements of CH₃Br at Gosan for 2008–2019. The box encloses the interquartile range (IQR) defined at 25–75 percentiles, and whiskers represent maximum (top) and minimum (bottom) enhancements. The solid and dot lines in the boxes represent the mean and median value of the data, respectively.**




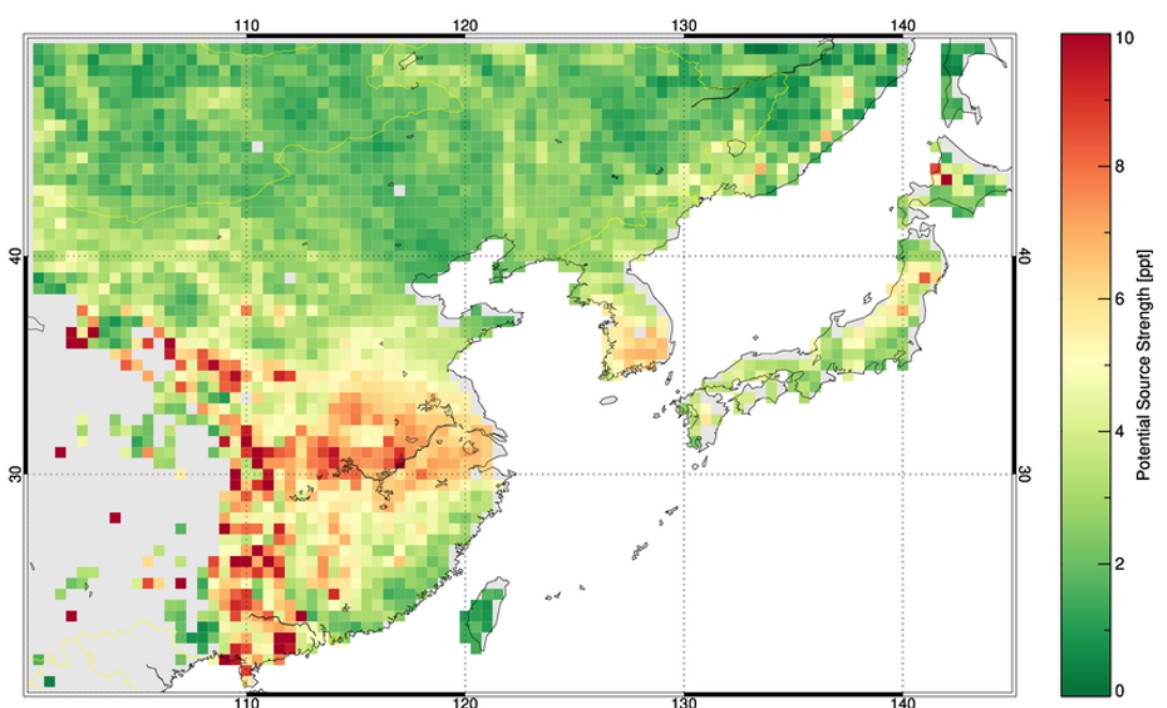

**Figure 5: Potential source regions for CH₃Br emissions derived by back-trajectory analyses of enhanced CH₃Br concentrations measured at Gosan station from 2008 to 2019.**


**Figure 6: Observed concentrations of CH₃Br, CFC-11, benzene, toluene, and ethane at Gosan during May 2010; note the highly correlated pollution events between CH₃Br and CFC-11, also the concentrations of VOCs (benzene, toluene, and ethane), which are likely related to the biomass burning and general anthropogenic combustion processes increased simultaneously.**

625   .

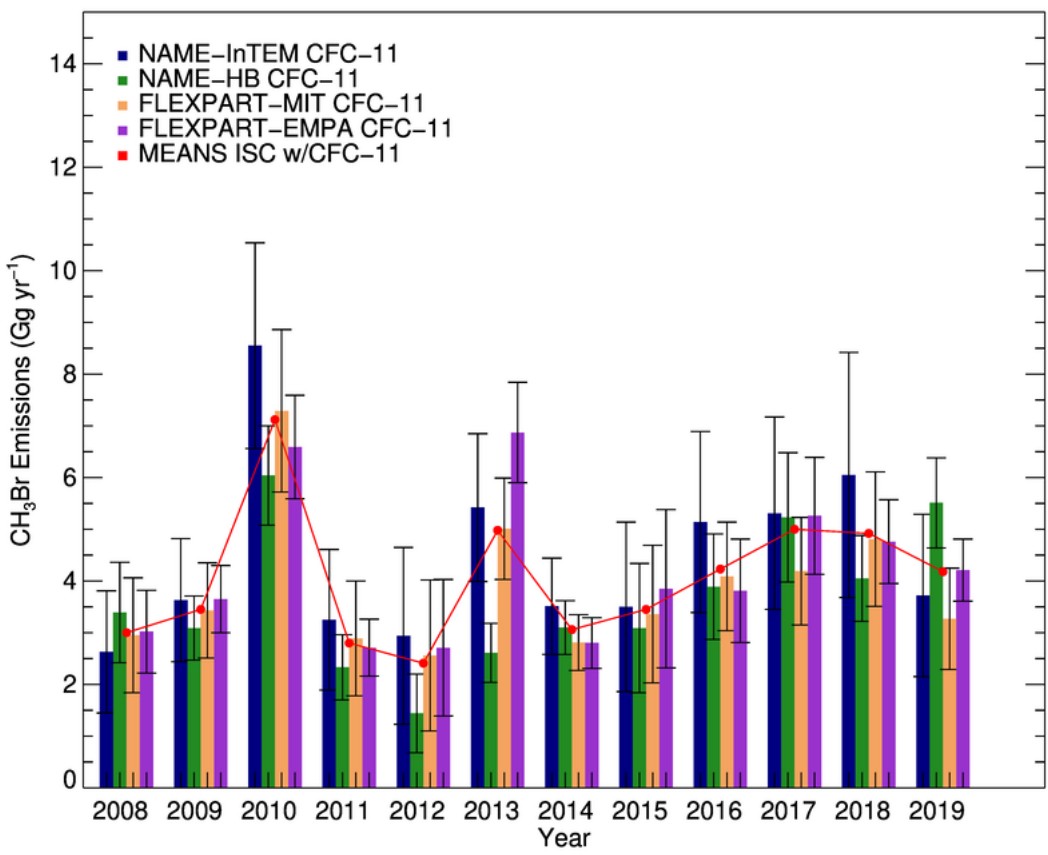

**Figure 7: CH₃Br emission estimates derived for eastern China by ISC from the observation data of CH$_3$Br and CFC-11 at Gosan during 2008–2019. CFC-11 emissions were taken from Park et al. (2021) and are estimated using four-independent inverse model frameworks (NAME-HB, NAME-InTEM, FLEXPART-MIT and FLEXPART-Empa). The bar plot of each color denotes the emission of CH$_3$Br with 1 sigma uncertainty derived from each inversion of CFC-11, and the average of the four different inversions is shown in red.**






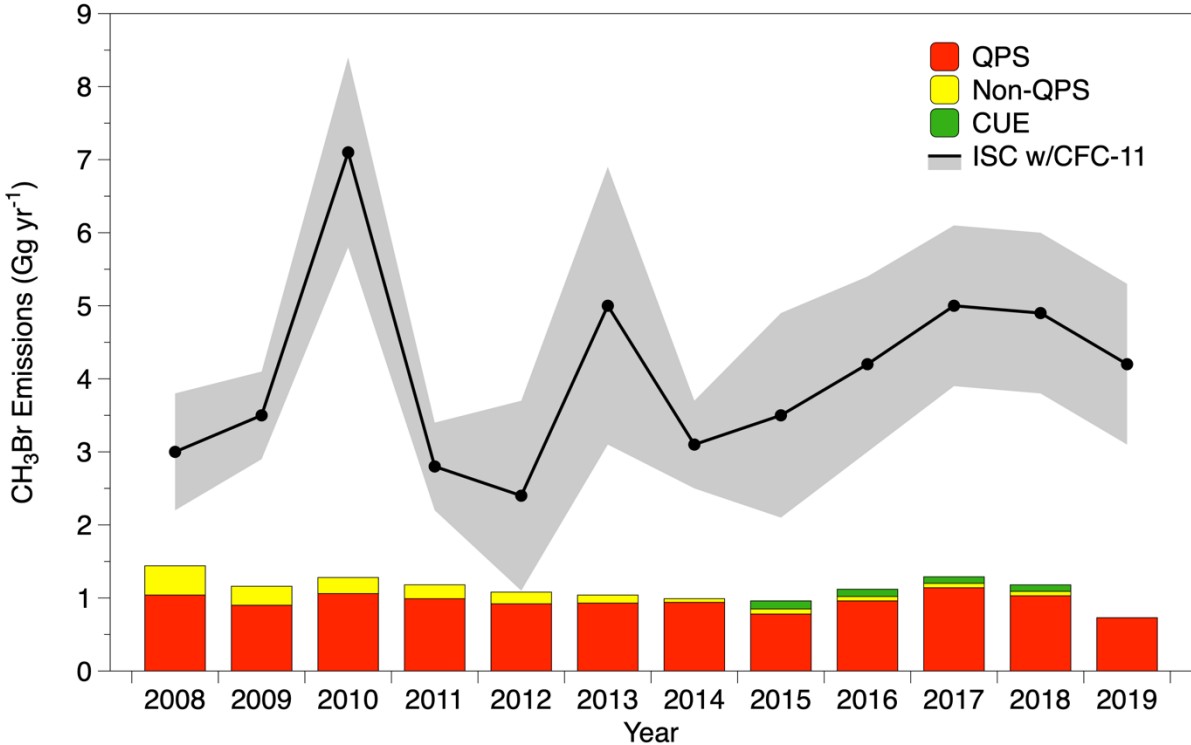

**Figure 8: Top-down emissions of CH₃Br for eastern China estimated by ISC using CFC-11 as the reference tracer (Figure 7). The bottom-up emissions of CH₃Br for China are based on the reported consumption data to UNEP for quarantine/pre-shipment (QPS), non-QPS and critical use exemption (CUE) categories.**



**Table 1: Annual means and standard deviations of CH₃Br baseline data and observed pollution signals at Gosan 2008 to 2019.**

| Year | Baseline | | Pollution | |
|---|---|---|---|---|
| | $CH_3Br$ (ppt) | Number of data | $CH_3Br$ (ppt) | Number of data |
| 2008 | $8.5 \pm 0.8$ | 933 | $13.1 \pm 5.1$ | 1402 |
| 2009 | $8.7 \pm 0.5$ | 955 | $13.0 \pm 4.6$ | 1187 |
| 2010 | $8.6 \pm 0.8$ | 1215 | $13.2 \pm 4.9$ | 1485 |
| 2011 | $8.4 \pm 0.7$ | 938 | $13.4 \pm 6.2$ | 1395 |
| 2012 | $8.4 \pm 0.6$ | 1069 | $12.5 \pm 4.3$ | 1360 |
| 2013 | $7.9 \pm 0.9$ | 1065 | $12.7 \pm 4.4$ | 1831 |
| 2014 | $7.9 \pm 0.6$ | 967 | $12.5 \pm 4.7$ | 1715 |
| 2015 | $7.8 \pm 0.7$ | 497 | $12.6 \pm 4.6$ | 1263 |
| 2016 | $8.0 \pm 0.7$ | 445 | $13.4 \pm 4.4$ | 688 |
| 2017 | $7.8 \pm 0.9$ | 425 | $11.7 \pm 4.0$ | 582 |
| 2018 | $7.0 \pm 0.4$ | 691 | $10.9 \pm 3.9$ | 899 |
| 2019 | $7.4 \pm 0.6$ | 1105 | $12.1 \pm 5.0$ | 1655 |




**Table 2: Bottom-up and top-down emissions of CH₃Br from 2008 to 2019 as presented in Figure 8. The bottom-up emissions are the sum of QPS, non-QPS and CUE consumption as reported to UNEP for all of China (data available at Ozone Secretariat website, http://ozone.unep.org). The top-down emissions were derived for eastern China by ISC method with CFC-11 as the reference tracer.**

| Year | UNEP reported ($Gg\ yr^{-1}$) | | | | ISC (CFC-11 ref.) |
| --- | --- | --- | --- | --- | --- |
| | QPS | Non-QPS | CUE | Total | ($Gg\ yr^{-1}$) |
| 2008 | 1.04 | 0.40 | - | 1.44 | $3.0 \pm 0.8$ |
| 2009 | 0.90 | 0.26 | - | 1.16 | $3.5 \pm 0.6$ |
| 2010 | 1.06 | 0.22 | - | 1.28 | $7.1 \pm 1.3$ |
| 2011 | 0.99 | 0.19 | - | 1.18 | $2.8 \pm 0.6$ |
| 2012 | 0.92 | 0.16 | - | 1.08 | $2.4 \pm 1.3$ |
| 2013 | 0.93 | 0.11 | - | 1.04 | $5.0 \pm 1.9$ |
| 2014 | 0.94 | 0.05 | - | 0.99 | $3.1 \pm 0.6$ |
| 2015 | 0.78 | 0.07 | 0.11 | 0.96 | $3.5 \pm 1.4$ |
| 2016 | 0.96 | 0.06 | 0.10 | 1.12 | $4.2 \pm 1.2$ |
| 2017 | 1.14 | 0.06 | 0.09 | 1.29 | $5.0 \pm 1.1$ |
| 2018 | 1.03 | 0.06 | 0.09 | 1.18 | $4.9 \pm 1.1$ |
| 2019 | 0.73 | 0 | - | 0.73 | $4.2 \pm 1.1$ |

**QPS: quarantine/pre-shipment, CUE: critical use exemptions, UNEP: United Nations Environment Programme**
