# Peer review of "Top-down and bottom-up estimates of anthropogenic methyl bromide emissions from eastern China"

_Atmospheric Chemistry and Physics, 2021_

## Author Comment (AC1)

**Response to All reviewers**

We thank the referees for their thoughtful and thorough reviews. We are pleased that all the reviewers see our manuscript as a valuable contribution to the field. We have made changes to the manuscript to answer the suggestions of the reviewers and clarified a few points raised in the review. We respond first to two main common comments of the referees below and all others were replied to under each referee's comment. A revised version of the manuscript including most of the changes suggested by the reviewers will be submitted to the editor. We thank the reviewers and the editor for their time and effort. [In the following, Reviewers' comments are in black and our responses are in red]

**Common Response #1**
**Robustness of the ISC linear regression of $\Delta$CH3Br vs. $\Delta$CFC-11.**

A common comment about the linear regression in the ISC method was that regression fits would largely depend on high-concentration data points, which might be outliers. While the observed scattering in the ISC plots is unavoidable because, for even well-correlated compounds, their emission sources, locations, and emissive patterns cannot be identical, we also questioned about the potential biases of the regression associated with the scattered data points and examined them by applying various fitting methods as described in the SI. For the high-value outliers, we tested the effect of filter-outed outliers on the analysis results, as the reviewers suggested. First, we compared the original regression result using all data (black) and the regression for the data with outliers removed (yellow) in Fig. R1a. To filter out outliers, we selected data in the range of Q1-1.5*IQR < the difference of CH3Br and CFC-11 < Q3+1.5*IQR (outliers removed). The WDR regression fits for the two datasets (with vs. without outliers) were not significantly different. Of sure, the fit to the filtered data showed a slightly better correlation coefficient (R).

Fig.R1b shows the revised CH3Br emissions estimated using the annual slopes (in yellow) and uncertainties derived from Fig.R1a. Their differences from the original emission estimates are negligible falling within their uncertainties.

Therefore, since our ISC analysis using robust WDR doesn't seem to be significantly affected by data outliers, we think it would be reasonable to present the emission results derived based on all available observation data without filtering.

We agree with the suggestion of the reviewers that Fig. S4 is better to be shown in the main text. So, we have modified as including annual regression slope values, uniform axis scales for every subplot, and larger font size, and then moved it into the main text. (in Fig.7). Fig.R1a show below is newly added in the SI to represent the outlier analysis.

[Figure]

**Fig. R1a** Annual correlations between the enhanced concentrations of CH₃Br and CFC-11 above the baseline observed at Gosan from 2008 to 2019. The weighted Deming regressions were applied for all data (which are represented in Fig. 7 of the main text; black) and for outlier-filtered data (yellow). The yellow asterisks correspond to observation data points that were removed as potential outliers.

[Figure]

**Fig. R1b** Emission estimates of anthropogenic CH3Br derived based on the ISC regression slopes and related uncertainties for all data (black) and outlier-filtered data (red).

**Common Response #2**

**Consideration of individual ratios between ΔCH3Br and ΔCFC-11.**

The potential bias associated with high-value outliers in determining ISC regression slopes were examined in Common Response #1 by applying an outlier filter, as suggested by the reviewers. Another way that can be considered in making a correlation between ΔCH3Br and ΔCFC-11 is to take ratios of individual CH3Br enhancement data points against corresponding CFC-11 enhancement data and determine an annual ratio by averaging all the ratio values for a given year. Since it would be a meaningful alternative to the ISC regression method as a reviewer suggested, we tested the individual ratio method. Note, however, that the ratio approach would be less robust because the ratio values for even poorly-correlated, low enhancement data sets are equally weighted with those taken highly-correlated ΔCH3Br vs. ΔCFC-11 sets. Fig. R2 shows the annual averages and 1-σ standard deviations of the enhancement ratios of individual CH3Br and CFC-11 datasets in comparison with the ISC regression slopes for all data and outlier-filtered data. The outlier filtering does not make any statistical difference from the original regression as discussed in the common response #1, and also the averages of the individual ratios are basically identical to the regression slopes, but with larger uncertainty ranges. For this reason, our WDR slopes seem to be robust enough to show a representative correlation for all observed enhancement data between CH3Br and CFC-11, so we decide to keep the original regression analysis and results in the main text without additional discussions about outlier filtering and the ratio approach to avoid unnecessary complexity.

[Figure]

**Fig. R2** Comparison of annual averages of individual ratio values (i.e., ΔCH3Br/ΔCFC-11) with annual regression slopes for all data and outlier-filtered data. Error bars denote 1-σ standard deviation ranges.

**Common Response #3**

**Why CH3Br emissions were estimated only from eastern China.**

As shown in the analysis of Air-mass classification and PSS, both China (especially eastern China) and South Korea appear to have an impact on the increments in the observed CH3Br mole fractions at Gosan. However, in this study, we focused on the anthropogenic CH3Br emissions only from eastern China for the following reasons.

1) Eastern China is certainly the region with the highest national-scale emission in East Asia when compared with that of South Korea or Japan.

2) Currently, we don't have any inverse model-based regional emission estimates for CH3Br in east Asia mainly because of some modeling challenges regarding natural emission sources. In this sense, our ISC method could be a relatively simple and accurate approach to estimate regional emissions for anthropogenic CH3Br, as stated in the main text. However, the ISC approach relies on a correlation with a reference tracer compound, of which the regional or national-scale emissions should be quantitatively well-defined. We have a well-correlated CFC-11 reference; its regional emissions were very well quantified up to the year 2020, and its representative anthropogenic sources were also well defined in eastern China. However, recent emissions of CFC-11 from South Korea and Japan were negligible compared to those from eastern China and thus its correlation with CH3Br were very poor. So, if we would find a relevant anthropogenic reference substance to be able to apply a similar CH3Br ISC analysis for South Korea and Japan, we certainly extend to our emission estimates to other countries.

**General major comments**

Is there a particular reason why the focus was set on Chinese CH3Br emissions? By looking at Figure S2b it appears that (despite a 'proximity' bias) Korean emissions are on a similar order of magnitude. Also, by comparing Figure 4, where maximum enhancements are around 10-13 ppt, and Figure 2, where there are many much larger pollution events, it suggests that other large emitters are nearby.

→ Yes, definitely correct as you said, eastern China as well as South Korea have potential signals that cannot be ignored. However, the total amount of anthropogenic CH3Br emissions in eastern China is higher than in other regions when the gross area is considered. And, emissions from the reference tracer (CFC-11) used in the ISC method for regional emission estimation are well defined for the East China region but not for South Korea. Therefore, we focused on eastern China in this study.

Please, refer the common response # 3.

My main concern is about why the authors can exclude oceanic algae and marshland production of CH3Br from being an important contributor in the observations at Gosan?

How conservative is the CH3Br concentration in air advected over the oceans?

Is algae CH3Br production in the waters around Gosan and between China and Gosan less than what one would e.g. assume for Mace Head?

For example, Fig 6, intended to show correlation of CH3Br with CFC-11 and some other tracers of both anthropogenic/natural origin, show such nice correlation for 20 May. However, for the period of 28-30 May, large pollution events of CH3Br are not matched with enhancements of CFC-11, benzene, toluene, ethane, and hence are not originating from biomass burning or general anthropogenic activities. Could this be an example of oceanic origin of CH3Br?

I suggest to produce Fig 5 (map with potential source regions) in a way that allows sources in the ocean.

→ You have pointed out an important point. Practically, natural sources of CH3Br also influence the mole fractions of CH3Br observed at Gosan. As you suggested, oceanic algae and saltmarsh are also one of factors and should be included in the total atmospheric CH3Br emissions (As mentioned in introduction). This research is focused on anthropogenic CH3Br sources. Natural sources, such as emissions from ocean, should be studied separately. For this reason, in this study, only anthropogenic emissions of CH3Br were estimated using anthropogenic tracers, excluding natural sources.

→ As shown in Fig. 6, the case where the CH3Br mole fraction increased during the period from May 28 to 30, 2010, but other anthropogenic substances did not increase, is most likely due to a natural

emission source. In the ISC method for estimating anthropogenic emissions, we only used cases where enhancement occurred at the same time as CH3Br and CFC-11.

Because the explanation for this was insufficient, the content was reinforced and described in the main text. Thanks for picking up the points.

→ So, we think that it would be better to represent only the land results excluding the ocean for PSS in Figure even we have PSS values over the ocean (to avoid adding confusion).

I question the robustness of the ISC correlations (shown in Fig. S4). The data look strongly biased towards a few large concentration enhancements. What happens if e.g. the highest 10-20 points each are removed? Many of the data plots in S4 suggest that many of the data points in the high concentration range are rather scattered (as suggested on line 246) but some suggest distinctly different branches. The data suggest that factors other than population activity (CFC-11 emissions) seem to contribute to some of these. If the high-CH3Br points had a natural component, and if these were removed, it could significantly reduce the anthropogenic emissions. Have the authors tried to apply another filter to understand if some of these branches are biased towards other sources (rapeseed, biomass, oceanic)? For example, a filter could be applied by looking at correlations of CH3Br to another substance, e.g. CFC-12. If the high-CH3Br samples stick out again compared to CFC-12, then that could be an indication of a natural source.

→ Yes, that's right. Obviously, the regions of all emission sources may not always be the same between tracers. Thus, we performed ISC method by applying the WDR method to derive a slope that well reflects the overall scatter trend, including these biases.

→ As you commented, we additionally compared CFC-12, another representative anthropogenic source, to see if it could be biased by natural emission sources (e.g., High CH3Br with Low CFC-11) in the scatters of CFC-11 vs. CH3Br. Fig. R3 shows annual scatter plots for cases of CH3Br, CFC-11, and CFC-12 enhancement occurred simultaneously. Looking at the scattered trend of CFC-12, in general, CFC-12 increases together with CH3Br and CFC-11 simultaneously, and the specially tilted regions (High CH3Br with Low CFC-12 and CFC-11) that can be regarded as natural sources, are not classified. This indicated that the enhancement data used in ISC originated from an anthropogenic source of emissions. Similar to CFC-11, we can notice that the relationship between CFC-12 and CH3Br also showed a good correlation (except for after 2018 and 2019, when hardly emitted CFC-12). However, we adopted CFC-11 because we need annual emissions from a well-established reference tracer for eastern China.

→ As mentioned in the above response, we only used the case where the enhancement of CH3Br and CFC-11 occurred simultaneously in the ISC method in order to minimize the enhancement of CH3Br caused by these natural sources and to see only the anthropogenic effect. These reasons have been added

to the manuscript and described in detail. Also, refer to common response #1 for ISC analysis of outliers that exist, please.

[Figure]

**Fig. R3** Annual correlation between the enhancement of CH3Br and CFC-11 above baseline measured at Gosan from 2008 to 2019. The color labels for each scatter point denote the enhancement mole fraction of CFC-12. All points correspond when the enhancement of CH3Br, CFC-11, and CFC-12 occurred at the same time. Note that, in 2018 and 2019, enhancement of CFC-12 hardly occurred, so the number of co-matched data was very small. The red solid line denotes the regression slopes between CH3Br and CFC-11 as used in ISC.

Given the large dynamics of CFC-11 emissions from this region over the past years, an inspection of time-records of yearly (or monthly) CFC-11 emissions and CH3Br/CFC-11 enhancement ratios (from Fig. S4) would be very informative and could be added to Fig. S3.

→ Revised. We have added a graph of the mean of CFC-11 emissions used in this study to make it easier for readers to understand. Is the meaning of 'CH3Br/CFC-11 enhancement ratios' correct as of the average of each enhancement ratio? If my understanding is correct, as other reviewers have also suggested, the enhancement ratio of CH3Br/CFC-11 can be good information. However, there is a concern that taking the enhancement ratio gives all the weights for each observation the same (whether the enhancement mole fraction is high or low). As described in common response #2, the difference between the regression slope and the ratio is small, and we think that it would be better not to indicate the ratio because it may cause confusion with the regression slope that is used to estimate emissions in this study.

Using Mace Head and Cape Grim as reference background station: Is Mace Head a good choice for a NH background extraction given its large local oceanic sources? Is the pollution filter working well for a station with presumably high local sources?

→ If my understanding of the Pollution filter you mentioned is correct, the classifying method of baseline and pollution which determine the background concentration within the temporal window of 120 days has been generally applied in AGAGE networks (O'Doherty et al., 2001).

→ Background concentrations may also potentially include to some extent an increase due to continuously occurring contaminants. That's why we track the regional emission using enhancement mole fractions.

→ Mace Head and Cape Grim are sites that traditionally represent the Northern and Southern Hemisphere to monitor the remote background atmospheres. They are good for judging the precision and seasonal variability of the baseline CH3Br mole fractions observed at Gosan. We have described this in the main text with a reference (Prinn et al., 2018).

Mentioning of SO2F2

In my view, the authors miss a chance to strengthen their CH3Br study by not including a similar analysis for SO2F2. It would be a strong plausibility test of the CH3Br results and help in the interpretations. If the decline of the CH3Br emissions from reported consumption (1.44 Gg to 0.73 Gg) from 2008 to 2019 is not at least partially matched by a similar-magnitude increase of SO2F2 (assuming insignificant use of other, presumably more expensive alternatives), then this could be supporting the conclusions of this work. I don't understand the logic behind the ISC of CH3Br vs SO2F2 (l. 372 ff, Figure S8, Table S3), if one is a replacement for the other, then why should they correlate? Why was the analysis not done of SO2F2 vs CFC-11? Also, I am confused about the statement of a remaining discrepancy of 3.5 Gg/yr (l. 371 and 373) when before the discussion was about a discrepancy of 3 Gg/yr.

Is the alternative SO2F2 really that much more expensive and less effective? I am just surprised that the Chinese authorities would tolerate another forbidden use of a MP substance after they were caught with CFC-11.

→ SO2F2 is not used as an alternative for all of the CH3Br usage. As mentioned in the main text (with Gressent et al., 2021), in eastern China, almost usage of SO2F2 is being for post-harvest treatment.

→ Analysis of CFC-11 vs SO2F2 was also performed in advance during the research process as shown in Fig. R3. The relationship between these two substances showed a relatively poor correlation compared to those of CFC-11 vs CH3Br, because the use of SO2F2 is almost post-harvest treatment in eastern China (unlike CH3Br, which is used for various QPS applications). Thus, we estimated only the emission contribution of CH3Br for post-harvest purposes usage from the good correlation between

CH3Br and SO2F2. (It seems that CH3Br and SO2F2 use source was colocated, thus they are not completely replaced and co-emitted. This was written in addition to the text.)

→ We additionally describe the values of the CH3Br emissions for post-harvest treatment which estimated by ISC method with SO2F2 in Table S3.

→ It is the rest of the emissions subtracted contributions by rapeseed and biomass burning of agricultural residues, not for fumigation purposes from estimated total 4.1 Gg/yr. By specifying this precisely in the manuscript, we have eliminated the confusion.

[Figure]

**Fig. R4** Annual correlation between the enhancement of SO2F2 and CFC-11 above baseline measured at Gosan from 2008 to 2019.

Minor comments

Abstract:

l. 26: I am having difficulties to understand reproduce the value of -0.13 ppt/yr from the decline from 8.5 to 7.4 ppt over the course of the eleven years.

→ The linear trend for all baseline mole fractions for the entire period from 2008 to 2019 as shown in below Fig. R5 (red solid line).

[Figure]

**Fig. R5** Linear trends for the baseline mole fraction and the whole period observed at Gosan from 2008 to 2019.

l. 29: I suggest to extend 'estimate anthropogenic ...' to 'estimate mean anthropogenic ..'.
Please make clear if the +- 1.3 Gg/yr is simply the variability in the yearly estimates, or if this includes some uncertainty estimate.

→ Revised. We modified as 'an average of 4.1 ± 1.3 Gg yr-1 in 2008–2019'.

l. 32: Why the term 'largely'? Is this word an expression of quantiy or uncertainness in the origin of the discrepancy?

→ Revised. It was an expression of a quantitative difference. Largely is a relative expression, so deleted it.

l. 51 'reduced completely' is an expression that doesn't make sense.

→ Revised. The 'completely' is an extreme expression, so it changed to 'mostly'.

l. 53 'reduction of 60'000 tonnes' Over what time frame? Consider using same units as later in the text (Gg).

→ Revised. We have specified for the period (from 1998 to 2017) with added a reference (MBTOC, 2018). 'tonnes' is used as the unit of data reported in the Ozone Secretariat, and since it is also expressed in 'tonnes' in the MBTOC, we intend to keep it at 60,000 tonnes instead of 60 Gg.

l. 55 '.. due to the phase-out of other uses ..' This is a confusing part of the sentence. Is it necessary?

→ This means that QPS uses now account for 98% of total consumption due to the phase-out of other regulated uses. we modified 'other uses' to 'other regulated uses' in more detail.

l. 57 'As a consequence ..' This statement assumes that natural sources of CH3Br have remained constant over this time frame, which seems rather speculative.

→ This is suggested in the Scientific Assessment of Ozone Department 2018 (Engel and Rigby et al., 2019), and although it is not assumed that the natural source remains constant, the main cause of the decrease in atmospheric background mole fractions is the influence of phase-out on controlled uses.

l. 58: Define 'ppt' the first time used in the text

→ Revised, we mentioned it as 'parts per trillion'

l. 62, natural emissions: Is there an estimate of the total natural CH3Br emissions from all these studies in the literature? If so, could you mention it? What fraction could potentially be assigned to the region of interest'

→ In the references, the global emission values for these natural sources are estimated and presented, but the values are not presented for East China or East Asia, which are areas of interest in this study. Since this study mainly focuses on anthropogenic sources, we did not refer to the quantitative values for natural sources.

l. 77: It would be instructive to give a typical 'activity factor' for these applications.

→ Revised, we mentioned this in Introduction as mentioned in the main text (65% for reported non-QPS consumption and 84% for the reported QPS consumption).

l. 84: The word 'resultant' is confusing. Is it necessary?

→ Revised. It removed.

l. 114: Suggest to extend 'of most the Medusa...' to '... of most of the Medusa ...'.

→ Revised.

l. 123: Remove the first 'and'. What is similar to the annual cycles at Gosan and Mace Head -- the amplitude, phase.

→ Revised.

l. 125: See also abstract: How do the authors derive a decline of 0.13 ppt/yr from 8.5 ppt to 7.4 ppt in 11 years?

→ Same as previous answer. The linear trend for all baseline mole fractions for the entire period from 2008 to 2019 as shown in Fig. R4 (red solid line).

l. 127: 'data in 2011-2012': Why only this period? Does this suggest that in other years, it is not consistent, or are data missing, or is there another reason? please clarify.

→ Revised. In the Scientific Assessment of Ozone Depletion 2014 (Carpenter and Reimann et al., 2014) have provided growth rate values for 2011-2012. We have corrected the sentence in the main text to 'Reported period'.

l. 159: Specify which boundaries are meant (modeling boundaries, geographical boundaries, boundaries to what/where?)

→ Revised. It means the boundary of the estimated modeling result. we reflected in the manuscript.

l. 198. 19 and 21 May, which year?

→ Revised. in 2010.

l. 197 paragraph: The authors point out the good correlation between CH3Br and the tracers benzene, ethane, toluene, which have both biomass burning and also other anthropogenic sources. Based on that, wouldn't then CO be a good tracer for ISC, to capture the sum of biomass burning and anthropogenic sources??

→ Yes, from the correlation of benzene, ethane and toluene, we can infer that there is a combustion-related emission source. As mentioned in the text, CO is an excellent tracer that has been widely used in various previous studies. Unfortunately, we do not observe CO at Gosan. This is one of the reasons we chose CFC-11 as a reference tracer which can be observed simultaneously with CH3Br in Medusa.

l. 212: extend 'enhancements' to 'concentration enhancements'.

→ Revised, and we changed the expression of 'concentration' to 'mole fraction' as the suggestion of another reviewer.

l. 221: 'of the estimated CH3Br'. What are the units for the uncertainties? If unitless, then say so, if with units, then something is wrong here as SigmaE-CFC-11 would need to have the same units as the Sigma-alpha.

→ The unit for uncertainty is same as that of CH3Br emissions. The error propagation method is expressed in more detail. This is the same method used in Shao et al., 2011.

l. 231: Suggest to extend to '..the estimated emissions...'

→ Revised.

l. 241: suggest to extend to '... residual errors for both X and Y ...'

→ Revised.

l. 248: Suggest to change to '.. For most of the observation' (add 'the', remove 'entire')

→ Revised.

l. 249: The (e.g. R=0.7 in 2009) is not really a typical example (e.g.) but it is the best taken.

→ Yes. R=0.7 in 2009 was the best year. As your comments, we replaced it with R=0.48 in 2011, which is a typical value.

l. 249: is the < symbol correct, shouldn't it be >?

→ Yes. '>' is correct. It is revised.

l. 260: The statement of the emissions being 'relatively' constant is rather subjective.
Same for 'small fluctuations'. One could argue that year-to-year fluctuations are huge in two cases.

→ We have minimized the subjective expression as much as possible by deleting the expression 'small fluctuation of year to year' in the manuscript and additionally mentioned the emission range for the entire period.

l. 268, wildfires: If wildfires could be responsible for the 2010 and 2013 peaks, then what could wildfires be in other years. As far as I understand, these are not included in the bottom-up

estimates. The contribution of wildfires should be clarified semantically and quantitatively in the paragraph on biomass burning (301 ff).

→ As mentioned in the manuscript, the cause of the sharp increase in 2010 and 2013 is puzzling. We have determined that the wildfires in China due to El Niño have a slight potential to impact. However, the emission of wildfires cannot fully account for these large increases in emission, and hard to quantify the magnitude. We described in a little more detail for these points, so that it would not be misunderstood as if the rare wildfire completely explained this sudden peak. Also, we change the subtitle of ' biomass burning' that have comprehensive meaning to 'biomass burning of agricultural residues' in detail to avoid confusion, because most of the biomass burned in eastern China is mainly generated by agricultural residues (in common year, the contributions of wildfires from a complex mixed of natural and anthropogenic sources, have little impact in eastern China).

l. 278: Second last sentence. It is unclear, why this observation is particularly worth mentioning.
→ Revised. Remove that sentence.

l. 285: Suggest to change '... the actual difference in ...' to either '...the actual difference in...' or to ' .. actual differences in ...' (and then 'are' instead of 'is').
→ Revised. Change to 'the actual difference in'.

The paragraph on biomass burning (301ff) is confusing and needs improvement. Is the fraction 'agricultural open-field burning' (l. 303) the same as the 'agricultural waste' (l. 313)?
If this is the same, and, according to l. 316 turns out to be insignificant (0.07 Gg/yr compared to 3 Gg/yr for the difference between top-down and bottom-up) then why do the authors think this would be seen in a seasonality plot (Fig. S5)?
If 'biofuel' burning is a substantial contribution, then would one not expect peak $CH_3Br$ observations in winter?
Where are 'wildfires' included and where aren't they?
→ We changed the subtitle from Biomass burning to Biomass burning of agricultural residues.
→ Yes, agricultural waste is comprehensively included in agricultural open-field burning. It is thought that even not significant emissions may affect the amount of seasonal enhancement. Of course, it does not explain all the seasonal variations as shown in Fig S5. Therefore, we express it 'contribute slightly partly to the seasonality'.
→ As mentioned in the text, according to Zhang et al., 2020, it is reported that biomass burning in eastern China is mainly caused by crop residues. Of course, biofuels and wildfires cannot be dismissed, but the impact will be even more negligible than the burning of crop residues.

→ It is thought that even not significant emissions may affect the amount of seasonal enhancement. Of course, it does not explain all the seasonal variations as shown in Fig S5. Therefore, we express it 'contribute slightly partly to the seasonality'.

l. 314: 'from the field experiment'...? From which experiment.

Perhaps this is should say '... based on field experiments...' or '... based on a field experiment....'

l. 314: For the number '1.1 g tonnes-1'. Please be more specific, is this tonnes of fuel, tonnes of dry fuel?

It seems that the authors do not include CH3Br from fires other than agricultural and biofuel, i.e. wildfires. Are these negligible?

→ Revised, change to 'based on a field experiment'. Emission factors are given for dry matter burned. We added in the manuscript. As mentioned above, it is hard to separate the wildfire as natural sources and anthropogenic sources. According to Zhang et al., 2020, biomass burning in East China predominates by the burning of agricultural residues, so wildfire is judged to have little effect.

l. 318: I suggest to replace 'results' by 'resulted'.

→ Revised.

l. 346. Are there studies supporting the statement that CH3Br is more effective than e.g. SO2F2. And have the authors verified that CH3Br is indeed significantly cheaper than SO2F2. I am just surprised that the Chinese authorities would tolerate another forbidden use of a MP substance after they were caught with CFC-11.

→ Not all alternatives to CH3Br are less effective than CH3Br. To reduce confusion due to extreme expression, we express that sentence as the 'lowest cost effective'. (MBTOC, 2018)

l. 386: I suggest to replace 'In recent years, CH3Br accounts for ...' to 'In recent year, CH3Br has been accounting for ...'.

→ Revised.

l. 388: I suggest to state this a bit more carefully, e.g. 'if any potentially unreported ...'

→ Revised.

l. 395: I checked data availability. on /gc-ms-medusa there appears to be a data set with monthly mean results and one with high-resolution. The Gosan CH3Br data is only in the former, but should also be in the latter, after all, analysis was done on individual measurements (ISC).

→ High-frequency CH3Br data of Gosan have been uploaded to the AGAGE website.

Figure 4: Caption: Clarify if these are all data, or already filtered for a specific sector of origin. Maximum enhancements are in the range of 10-13 ppt, and when compared to Figure 2, it would suggest that the data in Fig 4 are already a selection. However, also comparing with Figure 2, it appears that large and frequent emissions are also derived from other regions. It re-iterates the question on why this paper only focusses on China, if presumably there are other large emitters.

→ These are for all observed mole fractions at Gosan for the entire period (unscreened for specific regions). In this text, it means the annual variation range of the mean value. (3.6~4.6 ppt).
→ This is a repeating answer, but the reason for focusing on Eastern China is that it corresponds to the region with the highest emission when considering the gross area, and a quantitatively well-defined reference tracer value is required to apply the ISC method. South Korea is also not a negligible amount of emissions, which we will continue to look at in the next phase of the study. Please, refer to common response #3 for details.

Figure 5: Does the back trajectory analysis put any CH3Br sources over the ocean? Where, how much? For a compound with both anthropogenic and natural sources, would it make sense to include these in the plot? Same comment for Fig. S6 for SO2F2. This could actually help to understand the differences in the oceanic/marshland contribution of CH3Br assuming that there are no sources of SO2F2. It would be usefult to mark the Gosan station on these maps.

→ The analysis of PSS was performed using enhancement data observed at Gosan, and it can be seen that the constant emissions from natural sources are included in the baseline concentration to some extent. (Of course, the enhancement value does not mean that events occurring in the ocean are not included at all). In the process of tracking the backward trajectory, it passes over the ocean. However, since we are only looking at anthropogenic effects, the PSS of the ocean is not shown in the figure to avoid confusion. The figure was modified by indicating the location of Gosan station on the map, and it was also mentioned in the caption.

The contribution of CH3Br from South Korea appears small on Fig 5 but the pollution events from Korea (Fig S2b) seem large. Is this apparent discrepancy fully explained by the proximity of South Korea to the Gosan station?

→ As shown in Figure 5 and mentioned in the main text, CH3Br emissions (especially South Korea) are not small. If we look at it in terms of the number of people per unit area or the number of people, it

will surely make a contribution that is as much as that of eastern China. Korea has relatively less emissions because the area of the territory is narrow. this is the reason why we pay more attention to eastern China, and tracking CH3Br emissions to South Korea is a task that should be challenged in the next study. Please, refer common response #3.

Figure 8: What uncertainties are included in the grey band? Only those of the CFC-11 emissions? I suggest to include all important uncertainties (CH3Br/CFC-11) slope ratio uncertainties, others? I am missing a short paragraph in the text with a discussion on the uncertainties, that allows the reader to understand where the largest uncertainties are expected from.

→ Yes, estimated emissions are also including slope uncertainties. The average of anthropogenic CH3Br emission derived from the four independent inversion frameworks means the solid line, and the standard deviation corresponds to the shaded region. This is additionally mentioned in the caption.

Table 1. The table should be self-explanatory, so clearly state whether these are concentrations above baseline or actual concentrations during pollution events. Also, explain if this is a data set filtered for a specific region. Re-check all captions if they allow understanding of the numbers and figures without the main text.

→ It is the actual observed mole fractions, and is not a screened data for a specific region. The caption was modified to mention the mean, standard deviation, and numbers of data for the results shown in Figure 2.

Table 2, caption: Clarify if (and which) activity factors were used to convert consumption to emissions for the bottom-up estimates.

→ Revised.

Summary and Conclusion

When mentioning additional contributions, please separate rapeseed and biomass burning, the latter is, according to the authors, insignificant.

→ Revised.

Supplement:

Figure S1: How does one have to interpret these figures. Does a dark green mean that there is high residence time? This would then suggest a long residence time over the oceans. As stated earlier, how can the authors exclude that potential oceanic sources have lead to the high CH3Br observed at Gosan?

→ Air mass backward trajectories with considering the particle dispersion effect. Of course, because the sea exists between the high mountains and Shanghai, the residence time occurs along the sea level, but this means a narrow trajectory of air masses that originate from the vicinity of Shanghai. In the case of a case that has passed through a large area, the residence time is spread widely. It is to reiterate, we do not rule out the influence of natural sources, such as the ocean, on the CH3Br mole fractions. we estimated contribution of anthropogenic CH3Br using CFC-11 (emissions from anthropogenic only; without natural sources).

Figure S2b: Legend: Does this correspond to the classification in Fig? What regions are 'East China'? What regions are 'Korea'? I suggest to move the lower y-axis limits to negative such that data points and tick marks don't interfer. 'Red' and 'Purple' can bearily be distinguished on printed copies, I suggest to change colors. The jargon 'enhanced concentration' should be avoided as it is not clearly understandable when not used together with the text. 'enhancement of CH3Br above background' (like described in S4) is much clearer.

→ The results are classified by Fig.S2a. The colors of Fig. S2a and S2b are redistributed so that they are similar. Also, considering that it is difficult to distinguish between red and purple as you suggested, we have changed them to Eastern China (red), China (blue), Korea (green), and others (gray). As explained in the Supplement text, Fig. S2 corresponds to the eastern China (15, 16, and 17) region, modified by 'mole fractions of CH3Br above baseline'.

Figure S3: See general comments. Also, this Fig S3 is only referred to in the main text, but should also have some reference in the supplement.

→ As your suggestion in the general comment, we added the annual average emissions of CFC-11 emission, which correspond to the reference tracer used in ISC method, as (c) in Fig. S3. For a detailed

analysis of the concentration or enhancement of CFC-11, it would be good to refer to Rigby et al., 2019 and Park et al., 2021 presented as a reference.

Figure S4: This is an important figure because it demonstrates the quality and limitations of the ISC method. I suggest to consider showing it in the main text (e.g. by replacing the current Fig. 6 of the main text, which seems far less important). The figure needs improvement. The tick labels, the x and y text labels, and the subfigure titles (years) are too small.

Are the scales all the same? I suggest to make uniform scales for all subplots as a comparison between the individual plots is far more important than a potential loss of resolution in some subplots.

label the red lines with the numeric values of the slopes. It would be rather informative to draw a (dashed) line that would correspond to the CH3Br/CFC-11 ratio, which one would obtain to match the bottom up emission estimate (still using CFC-11 from earlier studies). In the caption, re-iterate that these are filtered data based on air mass classification and that the selection is made for a specific region.

→ Revised. Figure S4 has been moved to be included in the text according to the opinions of all reviewers, and the figure has been modified. The result for ratio is not significantly different from the regression slope as written in the common response, and we want to show a robust regression result that takes into account the uncertainty of all observations. Please, refer common response #1 and #2.

Figure S5: State what the vertical bars mean.
→ Revised.

Table S2: Add references for the UNEP reported data.
→ Revised.

**Major Comments**

MC1: Inter Species Correlations: This is the part that I am struggling with the most. While this method can work well, and can provide robust results, the authors do not present a convincing analysis here. The large outliers in either $CH_3Br$ or F11 are dominating your regressions. To estimate emissions, I would expect that you would want representative ratios for these two compounds, and not slopes dominated by a few high pollution events.

There are ways to assess the consistency of the slopes with one of the simplest methods being to remove the largest 5, 10, and 15% of data (as an example) and run the regressions to see what the differences are. From your supplemental plots, my eye tells me you are going to get drastically different numbers for several years if you remove the outliers. However, I would suggest you consider not using regression slopes for this analysis.

Regional scale tracer-tracer ratios are typically represented by broad distributions and are decidedly non-gaussian. An alternative method is to simply ratio each enhancement and take the median value, and the uncertainty of the median, which is more robust to outliers than either the arithmetic mean or regression slopes (see Miller et al. 2012). You can further assess the individual ratios by either performing bootstrap (removal with replacement) Monte-Carlo simulations or manually removing the largest 5% or 10% of the ratios and taking the median of the reduced dataset.

→ Thank you for your suggestion of another approach. In the regression method, the correlation can be very heavily dominated by outliers. We applied robust WDR that can cover the overall scatter trend well, and it demonstrated that there was no significant difference between the regression results using all observation data and the with outliers removed. And we additionally described these points in the manuscript with refer the Delvin et al., 1975. Please, see the common response #1.

→ We agree that using each enhancement ratio is a good alternative (as several previous studies have done). In ratio approach, the same weight is given to each observation data to acquire the average (or median) ratio value. We showed that the results of the regression slope were all within the uncertainty range of the ratio results. As described in common response #2.

→ We strongly agree that the boost trap and Monte Carlo methods you suggested are effective methods. However, it seems that the number of available data is not enough for us to apply the Monte Carlo method to our data.

MC2: Organization: I suggest the authors significantly re-organize the paper. Sections 2 and 3 are so short as to not warrant stand-alone section headings. Furthermore, sections 4 and 5

both contain methodological descriptions that could be combined with sections 2 and 3 into a comprehensive "methods" section, to be relabeled as section 2. In particular, sections 5.1 and 5.3 contain a significant amount of method description that disrupts the flow of the results and makes re-finding results after an initial read difficult. I suggest that all methodological description be placed into a new section 2: methods.

→ As you suggested, the text has been reorganized. (Sections 2 and 3 are combined to form instrumentation and measurement data) The methodological parts of 5.1~5.3 have been reorganized into sub-section of Data and Analysis.

The introduction is one of the longest sections of the paper and contains a lot of information that is relevant to $CH_3Br$, but not specifically to the conclusions of this paper. I suggest carefully going through the intro and trimming it down.

→ As you suggested, we have trimmed down and reconstructed the overall flow of the Introduction. Overall, contents are included in the introduction such as the source and sink of CH3Br, anthropogenic usage and regulation by MP, and previous studies for top-down and bottom-up emissions, and the needs of this study to define the anthropogenic CH3Br emissions in East Asia.

MC3: Lack of support for possible causes, I.E. $SO_2F_2$: Medusa systems measure $SO_2F_2$, and the authors suggest that a slow transition to $SO_2F_2$ as a replacement for $CH_3Br$ is a possible reason for the continued high emissions they detect at Gosan. While the authors present the Gosan $SO_2F_2$ record, they do not do any further analysis than the potential source regions are similar (which is trivial given the nearly identical uses of both chemicals) and showing pollution events are concurrent in time. Given that $SO_2F_2$ is commonly used to replace $CH_3Br$, it would strengthen the paper to add an analysis of this compound to support the notion that China is not using $SO_2F_2$ and rather is choosing to violate the Montreal Protocol.

→ As stated in the text, SO2F2 is one of the substances as an alternative to CH3Br, and among various CH3Br uses in eastern China, SO2F2 is mostly used for post-harvest treatment (structural fumigation purposes emissions are zero) (Gressent et al., 2021). Therefore, from the similar emission spatial properties with CH3Br, it seems that the CH3Br for post-harvest treatment would be mix-used in the process of alternative as SO2F2. These parts have been added to the main text for clarity.

→Unreported or inaccurately reported emissions from fumigation usage were suggested as the main cause of the high emissions in eastern China detected at Gosan. The slow transition to alternatives, including SO2F2, is also contributing to this.

MC4: Several points in the paper are underdeveloped/lack explanation: Firstly, the authors focus in on eastern China, and then do not present analyses of any of the other source regions they show in figure 5. Why are Korean emissions not also assessed? Or Japan (even though they are likely to be low based on figure 5)? Based on Figure S2b, air masses from Korea seem equally as elevated in $CH_3Br$ as the air masses originating in China. Without these analyses, the support for the conclusion that emissions, and overall atmospheric burden, are almost entirely from eastern China is incomplete.

→ The cause of the high concentration of CH3Br observed at Gosan is not entirely limited to the eastern China, and as can be notice from Fig. 5, it is a region that South Korea cannot be ignored. In order to apply the method (ISC) used in this study, the emission of the reference compound for the same period and region must be well defined. This is one of the reasons of analyzing and focusing on the East China region in East Asia. Please, refer common response #3 for details. In this regard, we have additionally described the reasons in the main text.

**Minor Comments**

For entire paper: the term "concentration" is used throughout. Please change to "mole fraction" as the AGAGE data is published as mole fraction. See IUPAC Green Book for further reference (https://iupac.org/what-we-do/books/greenbook/)

→ Revised. Thank you for suggestion. Often, we tend to use 'concentration' and 'mole fraction' interchangeably. We have changed 'concentration' to 'mole fraction'.

L26: -0.13 ppt yr$^{-1}$ is not the number I get when I divide 1.1 by 12.

→ Linear trend for all baseline mole fractions for the entire period from 2008 to 2019 as shown in below (red solid line). Refer Fig. R4 in same response of other reviewers.

L118-119: Add countries for both sites in addition to Lat/lon. Additionally, I suggest adding a panel to Figure 1 with the globe showing the location of Gosan, Mace Head, and Cape Grim. This allows the reader to quickly see the spatial relationship between the three sites at a glance, rather than having to look it up.

→ Revised, we added the name of countries (Ireland's Mace Head and Australia for Cape Grim), And, we presented additional figure of the location of 3 sites to improve the understanding of spatial relationship.

Section 3: Add in justification for why Mace Head and Cape Grim are appropriate comparison sites to Gosan. Given CH$_3$Br's lifetime of about 9-10 months, I'd like to see more explanation of why CH$_3$Br at Cape Grim is included here.

→ Mace Head and Cape Grim are sites that traditionally represent the Northern and Southern Hemisphere to monitor the remote background atmospheres. They are good for judging the precision and seasonal variability of the baseline CH3Br mole fractions observed at Gosan. We have described this in the main text with a reference (Prinn et al., 2018).

L124: State which site the baseline mole fractions declined for.

→ Revised.

L138/139: State the dates explicitly for the periods of missing data.

→ Revised.

Figs 3 and 4: Given the missing data, the reader is left wondering how the authors have dealt with the lower number of samples during 2016, 2017 and 2018. For example, Figure 3 shows

monthly means for 3 year periods, yet based on figure 2, for 2017-2019, the months of March, April, and possibly May are really only 2019 data, as no data exist for those months for 2017 and 2018, at least from eye balling figure 2. See comment above about stating the periods of missing data explicitly.

→ Revised. We mentioned the period in manuscript.

L143: HYSPLIT citation is out of date per HYSPLIT website: see Stein et al., 2015 AMS: https://doi.org/10.1175/BAMS-D-14-00110.1.

→ Revised. We update the recent citation information (Stein et al., 2015)

L155 and section 4 overall: It is worth noting that while this section details and discusses the HYSPLIT back trajectories, at Line 155, the authors reference Figure S1, which denotes that FLEXPART was used. Please clarify what specific model you are using for this part of the analysis.

→ Revised. We specified in the text that Fig S1 additionally used FLEXPART to see the effect of dispersion instead of single trajectories of HYSPLIT.

L170 and figure S2a/b: It would be helpful if these two figures shared the same color scheme. I realize the colors are denoted in the legend, but in S2b, east China is bright red, wheras in S2a, bright red corresponds to the sea of South East. Korea is green in S2b and lumped into one category, yet neither North Korea nor South Korea are green in S2a. It is easier for the reader to quickly interpret the pollution events per zone in S2b using S2a if the colors match between figures.

→ Revised. As your suggestion, we reorganized the colors to improve the reader understand, so that it is easy to connect Fig.S2a and S2b.

L219 and 221: Equation 2: please subscript α, $E_{MB}$, and $E_{CFC-11}$. i.e $\sigma_{E-MB} = \sigma_{E-CFC-11} + \sigma_\alpha$. As it is written it is easy to mistake this as $\sigma * E_{MB}$, etc…

→ Revised. Eq. 2 has been rewritten by solving the formula in more detail, and E-CFC11 is indicated as a subscript so as not to be mistaken for multiplication.

Section 5.3: The term "significant" is well defined in statistics, and without supporting statistical analysis to show the significance of these regression slopes, I do not find the R values presented here convincingly significant. Additionally, the Pearson correlation coefficient R is known to be non-robust in the presence of outliers, and given the large outliers in your data,

the R values are almost certainly biased. See for example Devlin et al. 1975 (doi: 10.2307/2335508), or Zhou 1987 (doi: 10.1007/BF00897747).

Furthermore, figure S4 is somewhat integral to your paper and should be included in the main text, and not in the supplement.

→ Revised. we tone-downed 'significant' with remove that expression. ('show a significant correlation' to 'show a correlation'). We totally agree your suggestion, and moved the scatter plot of the results using all the data to the main text. Although the Pearson correlation coefficient R is non-robust when including outliers, our annual results of R for all data are commonly over 0.3 and good to enough (which required value via Millet et al., 2009). The Supplement provides additional information by showing the results of removing outliers, which shows that our robust regression method does not significantly depend on the presence or absence of outliers. Pleaser, refer the common response #1.

L355: This sentence has a typo or is missing a word after soil, I believe.

→ These are categories for Fumigant uses. (soil, commodity treatment, and structural). Not a typo or missing.

L359: same comment as above for L26.

→ Same as above response. Refer Fig. R4 in same response of other reviewers.

L369-374: This is confusing. You state there is a 3 Gg yr$^{-1}$ discrepancy and then state there is an additional 3.5 Gg yr$^{-1}$ discrepancy. What is this additional discrepancy?

→ It is the rest of the emissions subtracted contributions by rapeseed and biomass burning of agricultural residues, not for fumigation purposes from estimated total 4.1 Gg/yr. By specifying this precisely in the manuscript, we have eliminated the confusion.

L395: I went to download the data and while this link does provide access to the AGAGE Gosan station data, it a) does not include CH$_3$Br, and b) does not contain the data as presented in the paper (i.e) with the background filter applied. Please add the CH$_3$Br data at a minimum, and it would be nice to have this papers data set with all years included, and an additional column marking the samples as background and pollution as per the filter used in this paper.

→ High-frequency CH3Br data of Gosan have been uploaded to the AGAGE website.

---

## Referee Report (RR1)

**2nd Review of "Top-down and bottom-up estimates of anthropogenic methyl bromide emissions from eastern China" (Haklim Choi et al. 2021).**

Overall, the authors have done an excellent job re-working the paper and it is ready for publication with 1 minor exception.

**Minor point 1:** While I appreciate the authors taking into consideration the ratio method, they did not perform their analysis correctly. The revised analysis is done taking the average and standard deviation, and states that it is less robust, yet I specifically stated in my first review that it is the median, and not the mean, of the individual ratios that is more robust to outliers than the regression slopes. I've highlighted the points below:

Author response from Common Response 1:

*Since it would be a meaningful alternative to the ISC regression method as a reviewer suggested, we tested the individual ratio method. Note, however, that the ratio approach would be less robust because the ratio values for even poorly-correlated, low enhancement data sets are equally weighted with those taken highly- correlated ΔCH3Br vs. ΔCFC-11 sets. Fig. R2 shows the **annual averages and 1-s standard deviations of the enhancement ratios of individual CH3Br and CFC-11 datasets** in comparison with the ISC regression slopes for all data and outlier-filtered data.*

Initial review:

*Regional scale tracer-tracer ratios are typically represented by broad distributions and are decidedly non-gaussian. An alternative method is to simply ratio each enhancement and take the **median value, and the uncertainty of the median, which is more robust to outliers than either the arithmetic mean or regression slopes** (see Miller et al. 2012).*

The authors point out in their response to reviewer 2:

*In ratio approach, the same weight is given to each observation data to acquire the average (or median) ratio value.*

While it is true that both the average and the median give equal weight to each observation, it is not relevant because the mean and median are calculated in two different ways. To illustrate this, consider a made up 10-point data set of ratios:

[4,4,5,5,6,4,5,3,6,100]

This set has a mean of 14.2, but a median of 5.

Given that you are looking for a ratio that is representative of the bulk of your data, this shows how the median, while still giving equal weight to all values, is more representative of the bulk of the ratio data, and therefore more robust to outliers, than the mean (and regression slopes).

Therefore, I am not surprised by the authors revised analysis to see that the mean value of the ratios is not different than the regression slope method. However, the median of the ratios could be, and should be included.

The median annual ratio (not the mean) is easily calculated (since you have already done the bulk of the work getting the mean values) and then added to Table S1. Assuming this does not change the results in a meaningful way, after this is added the paper is ready for publication from my perspective. If there is a significant change, the results should be updated.

---

## Author Response (AR2)

**Response to Reviewer #2**

**Minor point 1:** While I appreciate the authors taking into consideration the ratio method, they did not perform their analysis correctly. The revised analysis is done taking the average and standard deviation, and states that it is less robust, yet I specifically stated in my first review that it is the median, and not the mean, of the individual ratios that is more robust to outliers than the regression slopes. I've highlighted the points below:

While it is true that both the average and the median give equal weight to each observation, it is not relevant because the mean and median are calculated in two different ways. To illustrate this, consider a made up 10-point data set of ratios: [4,4,5,5,6,4,5,3,6,100]
This set has a mean of 14.2, but a median of 5.
Given that you are looking for a ratio that is representative of the bulk of your data, this shows how the median, while still giving equal weight to all values, is more representative of the bulk of the ratio data, and therefore more robust to outliers, than the mean (and regression slopes). Therefore, I am not surprised by the authors revised analysis to see that the mean value of the ratios is not different than the regression slope method. However, the median of the ratios could be, and should be included.
The median annual ratio (not the mean) is easily calculated (since you have already done the bulk of the work getting the mean values) and then added to Table S1. Assuming this does not change the results in a meaningful way, after this is added the paper is ready for publication from my perspective. If there is a significant change, the results should be updated.

→ Thank you for pointing out again what we overlooked and did not present. We have no doubt that using the median of the proposed ratio is also a very robust emission estimation method. As shown in Figure R1 below, the result of our regression slope is much more similar to the median of the ratio values than the mean. (As the reviewer mentioned, the mean is definitely more sensitive to outliers). We added the annual median values and the $16^{th} \sim 84^{th}$ percentile range for all data in Table S1, and described the similarities in the main text with reference to Miller et al., 2012.

[Figure]

**Fig. R1** Comparison of annual means and medians of individual ratio values (i.e., ΔCH3Br/ΔCFC-11) with annual regression slopes for all data and outlier-filtered data. Error bars of means and medians denote 1-σ standard deviation ranges and $16^{th}$ to $84^{th}$ percentile ranges, respectively.